# On the Variance of Neural Network Training with respect to Test Sets and Distributions

**Keller Jordan**[*]
Independent researcher

## Abstract

Neural network trainings are stochastic, causing the performance of trained networks to vary across repeated runs of training. We contribute the following results towards understanding this variation. (1) Despite having significant variance on their test-*sets*, we demonstrate that standard CIFAR-10 and ImageNet trainings have little variance in their performance on the test-*distributions* from which their test-sets are sampled. (2) We introduce the *independent errors* assumption and show that it suffices to recover the structure and variance of the empirical accuracy distribution across repeated runs of training. (3) We prove that test-set variance is unavoidable given the observation that ensembles of identically trained networks are calibrated (Jiang et al., 2021), and demonstrate that the variance of binary classification trainings closely follows a simple formula based on the error rate and number of test examples. (4) We conduct preliminary studies of data augmentation, learning rate, finetuning instability and distribution-shift through the lens of variance between runs.

## 1 Introduction

Modern neural networks (Krizhevsky et al., 2012; He et al., 2016; Vaswani et al., 2017) are trained using stochastic gradient-based algorithms, involving randomized weight initialization, data/batch ordering, and data augmentations. Because of this stochasticity, each independent run of training produces a different network which may have better or worse performance than the average.

This variance between independent runs of training is often substantial. Picard (2021) shows that for a standard CIFAR-10 (Krizhevsky et al., 2009) training configuration, there exist random seeds which differ by 1.3% in terms of test-set accuracy. In comparison, the gap between the top two methods competing for state-of-the-art on CIFAR-10 has been less than 1% throughout the majority of the benchmark's lifetime[1]. Prior works therefore view this variance as an obstacle which impedes both comparisons between training configurations (Bouthillier et al., 2021; Picard, 2021) and training pipeline reproducibility (Bhojanapalli et al., 2021; Zhuang et al., 2022). To mitigate stochasticity, Zhuang et al. (2022) study deterministic tooling, Bhojanapalli et al. (2021) develop regularization methods, and many recent works (Wightman et al., 2021; Liu et al., 2022) report the average of validation metrics across multiple runs when comparing training configurations.

This paper contributes the following results towards understanding the structure and origin of variance in stochastic neural network training.

1. We demonstrate that random seeds which are "lucky" with respect to one split of test data perform no better than average on a second split. (Section 3.1)

2. We introduce the *independent errors* assumption, and show that it recovers the structure and quantitative variance of the empirical test-set accuracy distribution. (Section 3.2)

3. We present a formula for estimating the variance of the distribution-wise accuracy using only statistics observed on a finite-sample test-set. (Section 3.3)

4. We prove that the class-calibration property (Jiang et al., 2021) of neural network trainings implies finite-sample variance, and derive from it a formula which accurately predicts variance using only a single run of training for the binary classification case. (Section 3.4)

---

[*]Work performed while at Hive AI
[1]https://paperswithcode.com/sota/image-classification-on-cifar-10

We define the distribution-wise variance of a training algorithm to be the variance between runs of its performance on the test distribution from which the test-set is sampled. Although this quantity cannot be directly calculated from a finite test-set, we develop a formula (Equation 3) which provides an unbiased estimate of it using statistics collected on a test-set. Using the formula, we estimate that despite the fact that a standard CIFAR-10 training has a relatively large test-set standard deviation of 0.15%, its distribution-wise standard deviation is only 0.03%. We find similar results for a standard ImageNet (Deng et al., 2009) training (Section C). We conclude that the distribution-wise variance of standard neural network trainings is significantly less than their test-set variance.

To understand the reason why standard trainings have more variance on their finite test-sets than on their test-distributions, we turn to the class-calibration property discovered by Jiang et al. (2021). We first prove mathematically that although this property generates no constraint on the distribution-wise variance, it does imply a lower-bound the test-set variance (Theorem 4). We then empirically demonstrate that this lower bound closely matches the test-set variance across hundreds of binary classification trainings (Figure 6). This finding yields a tool for estimating variance without the need for multiple runs of training, which is more accurate than the commonly used binomial approximation (Dietterich, 1998; Raschka, 2018).

Our experiments and conclusions focus on standard CIFAR-10 and ImageNet trainings. We also investigate two exceptional scenarios where our conclusions do not fully apply: trainings where there is a distribution shift between the training and test distributions (Section C), and trainings with pathological instability (Section 4.1). Both of these cases have significant distribution-wise variance, unlike the standard training scenarios.

Finally, we conduct preliminary studies regarding the effect of learning rate (Section 4.3) and data augmentations (Section 4.2) on variance. When increasing the learning rate, accuracy begins to decline at the same point at which significant distribution-wise variance appears. Data augmentations significantly reduce variance.

## 1.1 RELATED WORK

A number of prior works investigate which sources of stochasticity are most responsible for variation between runs of training. Fort et al. (2019) observe that when using a below-optimal learning rate, randomized data ordering has a smaller impact than model initialization on the churn of predictions between runs. Bhojanapalli et al. (2021) similarly find that fixing the data ordering has no effect, while fixing the model initialization reduces churn. Bouthillier et al. (2021) report that data ordering has a larger impact than model initialization. Finally, Summers & Dinneen (2021) find instead that most variation can be attributed to the high sensitivity of the training process to initial conditions, with a single bit difference in starting parameters being sufficient to lead to the full quantity of disagreement between runs. We include a replication study of Summers & Dinneen (2021) in the appendix (Section D).

Dodge et al. (2020) study variation between runs of $BERT_{LARGE}$ finetuning, and achieve substantial gains to validation performance via the strategy of re-running finetuning many times and taking the best-performing result. We demonstrate (Section 4.1) that for the case of $BERT_{BASE}$, the low amount of genuine distribution-wise variance between runs indicates that any performance gains yielded by this strategy only amount to overfitting the validation set. On the other hand, for $BERT_{LARGE}$ we demonstrate that there is significant distribution-wise variance, supporting the use of multiple runs of training as Dodge et al. (2020) suggest. Mosbach et al. (2020) also study the finetuning instability of $BERT_{LARGE}$, and suggest to mitigate it by warming up the learning rate, training for longer with a smaller learning rate, and using bias correction for Adam (Kingma & Ba, 2014).

Many of our theoretical results draw upon the class-calibration property discovered by Jiang et al. (2021), who use it to provide a theoretical proof for their empirical observation that the rate of disagreement between two identically trained networks matches the error rate of each network. This builds on earlier works studying neural network ensembles, which reported that they are more well-calibrated (Lakshminarayanan et al., 2017; Nixon et al., 2020) and achieve higher accuracy under distribution shift (Ovadia et al., 2019) compared to individual networks. Also related is the observation of Mukhoti et al. (2021) that the usefulness of an ensemble's uncertainty scores depends upon the variance between the individual networks.

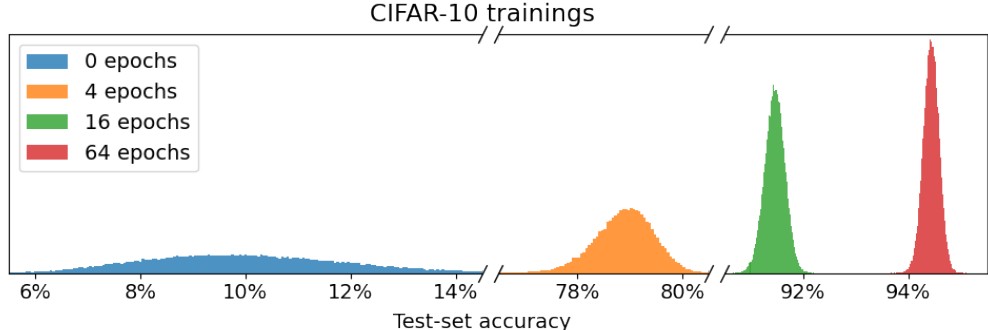

Figure 1: **Accuracy distributions.** The test-set accuracy distributions across our four training durations, displayed as unsmoothed histograms for 60,000 repeated runs of training each. The differences between the "luckiest" and most unlucky run (max minus min accuracy) are 13.2%, 6.6%, 1.7%, and 1.4% for the 0, 4, 16, and 64-epoch training durations, respectively. The standard deviations are 1.87%, 0.56%, 0.19%, and 0.15%.

Several prior works (Baldock et al., 2021; Ilyas et al., 2022; Lin et al., 2022) study the effect of randomly varying the data used to train a neural network, across a large number of training runs. Our study differs from these works in that we consider the simpler scenario of running a single training algorithm many times without varying anything except the training seed.

Broadly, our work is related to research aiming to understand the relationship between pairs of neural networks produced by repeated runs of training. This topic is of both theoretical and practical interest, and has been studied from a variety of angles, including the similarity of internal representations (Li et al., 2015; Kornblith et al., 2019), degree of correlation between predictions (Fort et al., 2019; Jiang et al., 2021), similarity of decision boundaries (Somepalli et al., 2022), path-connectivity in weight-space (Draxler et al., 2018; Garipov et al., 2018), linear mode connectivity (Frankle et al., 2020), and linear mode connectivity modulo permutation symmetries (Tatro et al., 2020; Entezari et al., 2021; Ainsworth et al., 2022; Jordan et al., 2022).

## 2 SETUP

**Notation.** This work studies neural networks trained to solve classification problems. We assume test-set examples are sampled independently from a distribution $\mathcal{D}$ over $\mathcal{X} \times \mathcal{Y}$ where $\mathcal{Y} = \{1, \ldots, k\}$ is the set of classes and $\mathcal{X}$ is the input space. When we refer to a training algorithm or configuration, we understand it to include everything necessary for training besides the random seed. That is, a training configuration already includes the choice of optimization algorithm, dataset, network architecture, and hyperparameters, so that only the random seed remains to determine the outcome of training. Let $\mathcal{H}$ be the hypothesis class of all functions of the form $h : \mathcal{X} \to \mathcal{Y}$. Following Jiang et al. (2021), for a stochastic training algorithm $\mathcal{A}$ we write $h \sim \mathcal{H}_{\mathcal{A}}$ to denote a hypothesis sampled from the distribution induced by the algorithm, *i.e.*, by running the algorithm and collecting the trained network. We write $\mathrm{err}_{x,y}(h) = 1\{h(x) \neq y\}$ to denote the condition that the neural network $h$ makes an error on the example $(x, y)$, so that $\mathbb{E}_{h \sim \mathcal{H}_{\mathcal{A}}}[\mathrm{err}_{x,y}(h)]$ is the proportion of runs of training which make an error on $(x, y)$. We additionally write $\mathrm{err}(h) = \mathbb{E}_{(x,y) \sim \mathcal{D}}[\mathrm{err}_{x,y}(h)]$ to denote the true error rate on the test distribution, so that the distribution-wise variance is $\mathrm{Var}_{h \sim \mathcal{H}_{\mathcal{A}}}(\mathrm{err}(h))$. For a test-set $S = ((x_1, y_1), \ldots, (x_n, y_n))$ we write $\mathrm{err}_S(h) = \frac{1}{n} \sum_{i=1}^{n} \mathrm{err}_{x_i,y_i}(h)$ to denote the test-set error. Finally, we write $\mathrm{err}(\mathcal{A}) = \mathbb{E}_{h \sim \mathcal{H}_{\mathcal{A}}}[\mathrm{err}(h)]$ to denote the average true error of the training algorithm.

**Core experimental setup.** For our core experiments (Section 3), we train ResNets on CIFAR-10. We study four different training durations: 0, 4, 16, and 64 epochs. The 0-epoch case corresponds to evaluating the network at initialization; this naturally has a random chance-level average accuracy of 10%, but some random initializations reach as high as 14% and as low as 6% accuracy. A complete description of each training configuration is provided in Section A. We execute each configuration 60,000 times and collect the resulting test-set predictions, yielding the accuracy distributions shown in Figure 1.

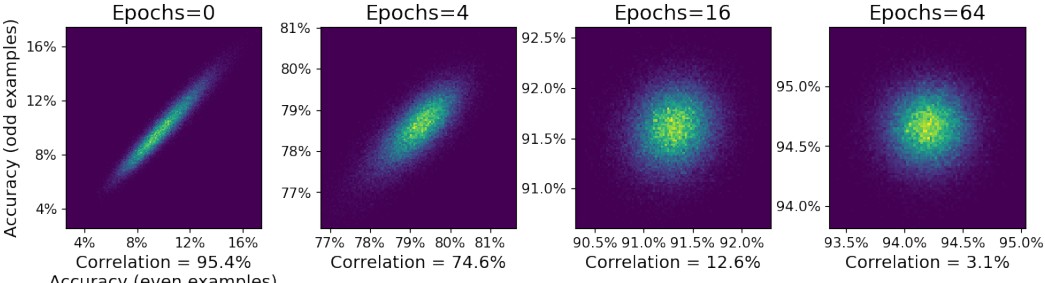

Figure 2: **Error rates on disjoint splits of test data become decorrelated when training to convergence.** We evaluate a large number of independently trained networks on two splits of the CIFAR-10 test-set. When under-training there is substantial correlation, so that a "lucky" run which over-performs on the first split is also likely to achieve higher-than-average accuracy on the second. As we increase the training duration, the two error rates decorrelate from each other.

## 3  DELVING INTO VARIANCE

### 3.1  DO LUCKY RANDOM SEEDS GENERALIZE?

In Figure 1 we observed that our standard CIFAR-10 training configuration has significant variation between runs. Even when training for a long duration, we found pairs of random seeds which produce trained networks whose test-set accuracy differs by more than 1%. In this section, we argue that this variance is merely a form of finite-sample noise caused by the limited size of the test-set, and does not imply almost any genuine fluctuation in the quality of the trained network.

Suppose we view the random seed as a training hyperparameter. Then we have observed that it can be effectively "tuned" to obtain improved performance on the test-set – on average, our training configuration attains an accuracy of $94.42\%$, but we found random seeds which reach above $95\%$, which is more than 10% fewer errors. However, this improvement on the test-set alone is not enough to conclude that the random seed genuinely affects model quality. What remains to be seen is whether these performance improvements can generalize to unseen data, or if we are merely over-fitting the random seed to the observed test-set.

To find out, we perform the following experiment. First, we split the CIFAR-10 test-set into two halves of 5,000 examples each. CIFAR-10 is already shuffled, so for convenience we simply use the odd and even-indexed examples as the two halves. We can view the first half as the hyperparameter-validation split and second as the held-out test split. Next, we execute many independent runs of training, with identical configurations other than the varying random seed. We measure the performance of each trained network on both splits of data. If lucky random seeds do generalize, then we should observe that runs of training which perform well on the first split also perform better than average on the second split.

To additionally determine the effect of training duration, we repeat this experiment for trainings of 0, 4, 16, and 64 epochs, using 60,000 independently trained networks for each duration. We view the results in Figure 2. For short trainings, the two splits are indeed highly correlated, such that runs which perform well on the first split also tend to do well on the second. But when training for longer, this correlation nearly disappears. When training for 64 epochs, for example, our highest-performing network on the first split does not even perform better than average on the second split. And on average, the top $1/4$ of runs with respect to the first split only perform 0.02% better than average on the second split.

This result has the following practical implication. Suppose we want to obtain a good CIFAR-10 model. Noticing significant variation between runs (Figure 1), we might be tempted to re-run training many times, in order to obtain networks with better test-set performance. However, according to Figure 2, this would be useless, because improvements on the test-set due to re-training will have near-zero correlation with improvements on unseen data. These networks would be "better" only in the sense of attaining higher test-set accuracy, but not in the sense of being more accurate on unseen data from the same distribution.

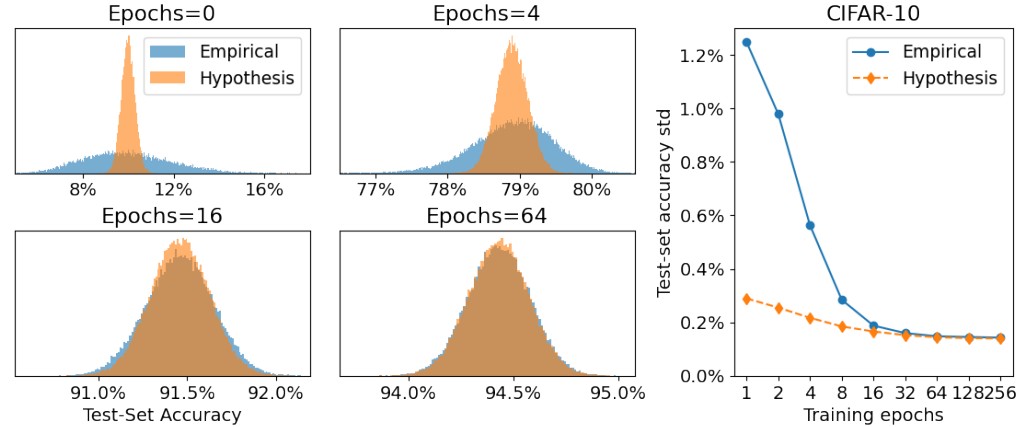

Figure 3: **Independent errors explain variance when training to convergence.** (Left:) We compare the empirical distribution of test-set accuracy with that generated by simulated an equal number of samples assuming the hypothesis of independent errors. The assumption is wrong for short trainings, but becomes a close fit as training progresses. (Right:) The assumption of independent errors accurately predicts the variance between runs of test-set accuracy when training to convergence.

## 3.2 INDEPENDENT ERRORS

In the previous section we showed that when training to convergence, disjoint splits of test data become nearly decorrelated, in the sense that networks which randomly perform well on one split do not perform better than average on another. We now formalize an even stronger condition, which states that independent runs of training make errors which vary independently on each example.

**Definition 1.** *The training algorithm $\mathcal{A}$ makes **independent errors** on a test-set if for every pair of test examples $(x_i, y_i)$ and $(x_j, y_j)$,*

$$\operatorname*{Cov}_{h \sim \mathcal{H}_{\mathcal{A}}} \left( \operatorname{err}_{x_i, y_i}(h), \operatorname{err}_{x_j, y_j}(h) \right) = 0 \qquad (1)$$

We demonstrate this condition in Figure 4. The error on each test example $(x_i, y_i)$, as a function of training stochasticity, is a Bernoulli variable with mean $\varepsilon_i := \mathbb{E}_h[\operatorname{err}_{x_i, y_i}(h)]$. The test-set error $\operatorname{err}_S(h)$ is the average of $n$ such Bernoulli variables. If $\mathcal{H}_{\mathcal{A}}$ makes independent errors, then each Bernoulli variable is independent, so that the overall test-set error must have variance equal to $\frac{1}{n^2} \sum_{i=1}^{n} \operatorname{Var}_h(\operatorname{err}_{x_i, y_i}(h))$. On the other hand, if the independent errors property is violated, then variance may be higher.

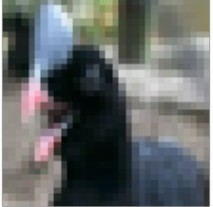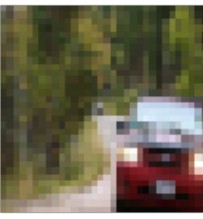

Figure 4: **A pair with independent errors.** (Left) is image 776 of the CIFAR-10 test-set. Out of 60,000 independent runs of 64-epoch training, 21,736 networks (36.2%) correctly predict this example. (Right) is image 796, which is correctly predicted by 36,392 networks (60.7%). The number of networks which predict both correctly at the same time is 13,103 (21.83%), which has a statistically insignificant difference to the quantity $0.362 \cdot 0.607 = 21.97\%$, which is the expectation if their errors are independent.

We next compare the distribution of accuracy-values that we observe in reality to what would be predicted by independent errors. Using our large collection of trained networks, we compute estimates of $\varepsilon_1, \ldots, \varepsilon_{10,000}$ for our four training configurations. From these estimates, using the assumption of independent errors, we both (a) compute predictions of test-set accuracy variance using the formula $\frac{1}{n^2} \sum_{i=1}^{n} \varepsilon_i(1 - \varepsilon_i)$, and (b) simulate samples of the test-set accuracy distribution. In Figure 3 we demonstrate that both of these become a close fit with reality as we train to convergence. For short trainings, the real accuracy distribution contains excess variance which is unexplained by independent errors, but for long trainings this excess disappears, so that the hypothesis becomes true in the aggregate sense of predicting the shape and standard deviation of the accuracy distribution. Additionally, in Figure 14 we find that only a small number of visually similar pairs violate the assumption of independent errors, and in Figure 13 we show that it compares favorably to the binomial assumption. In the next section we explore how small deviations from the independent errors assumption can be used to estimate variance with respect to the test-distribution.

### 3.3 ESTIMATING DISTRIBUTION-WISE VARIANCE

In Section 3.1 we showed that accuracy is decorrelated between disjoint splits of test-data, and argued that this implies small genuine variation in model quality between runs of training. In this section we clarify our notion of model quality, and present a method of directly estimating the true variance between runs, on the full test-distribution rather than a finite test-set.

Neural networks are typically evaluated by their performance on a test-set. However, what really matters is performance on the test-*distribution* $\mathcal{D}$, because this is what determines the expected performance on new batches of unseen data. Therefore, our notion of model quality is based on the distribution-wise error $\text{err}(h) = \mathbb{E}_{(x,y)\sim\mathcal{D}}[1\{h(x) \neq y\}]$, which we call the true error. The test-set is a finite sample from $\mathcal{D}$, so that test-set error is a noisy binomial approximation of true error.

Estimating the *mean* of true error across training stochasticity is relatively easy, because test-set accuracy is an unbiased estimator, as we have $\mathbb{E}_{S\sim\mathcal{D}^n}[\mathbb{E}_h[\text{err}_S(h)]] = \mathbb{E}_h[\text{err}(h)]$. Estimating the *variance* $\sigma^2 := \text{Var}_h(\text{err}(h))$ is more challenging, since the variance in test-set accuracy is an overestimate (proof in Section B.1).

**Theorem 1.** *In expectation, variance in test-set accuracy overestimates variance in true error.*

$$\mathop{\mathbb{E}}_{S\sim\mathcal{D}^n}\left[\mathop{\text{Var}}_{h\sim\mathcal{H}_\mathcal{A}}\left(\text{err}_S(h)\right)\right] \geq \mathop{\text{Var}}_{h\sim\mathcal{H}_\mathcal{A}}\left(\text{err}(h)\right) \quad (2)$$

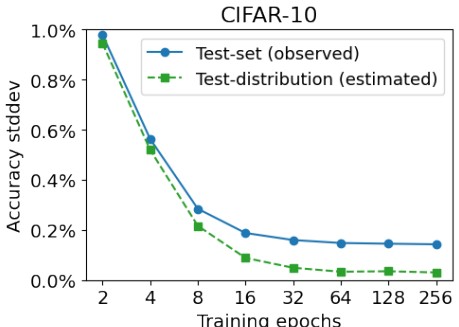

Figure 5: **Test-set variance overestimates true variance.** We use Equation 3 to estimate the true distribution-wise variance $\text{Var}_{h\sim\mathcal{H}_\mathcal{A}}(\text{err}(h))$. It becomes $22\times$ smaller than the test-set variance when training to convergence.

We investigate how to obtain an unbiased estimate of the true variance $\text{Var}_{h\sim\mathcal{H}_\mathcal{A}}(\text{err}(h))$. We recall the results of Section 3.2: When training to convergence, we found that test-set accuracy follows a distribution which can be approximately recovered by assuming that all test examples have independent errors (Definition 1, Figure 3). The distribution-wise variance in this case should be essentially zero. On the other hand, for shorter trainings we observed substantial test-set variance in excess of that predicted by Definition 1, *i.e.*, $\text{Var}_{h\sim\mathcal{H}_\mathcal{A}}(\text{err}_S(h)) \gg \frac{1}{n^2}\sum_{i=1}^n \text{Var}_h(\text{err}_{x_i,y_i}(h))$. For example, the hypothesis predicted that our 4-epoch configuration should have a standard deviation of 0.22%, but the observed value was much larger at around 0.56%. This suggests that these shorter trainings may have significant true variance between runs.

To formally connect the variance in true error to the excess in test-set variance over that predicted by the assumption of independent errors, we provide the following theorem (proof in Section B.2).

**Theorem 2.** *The following quantity is an unbiased estimator for true variance $\text{Var}_{h\sim\mathcal{H}_\mathcal{A}}(\text{err}(h))$.*

$$\hat{\sigma}_S^2 = \frac{n}{n-1}\left(\mathop{\text{Var}}_{h\sim\mathcal{H}_\mathcal{A}}\left(\text{err}_S(h)\right) - \frac{1}{n^2}\sum_{i=1}^n \mathop{\text{Var}}_{h\sim\mathcal{H}_\mathcal{A}}\left(\text{err}_{x_i,y_i}(h)\right)\right) \quad (3)$$

We calculate this estimate using many runs of training, with the fixed CIFAR-10 test-set. In Figure 5 we compare $\hat{\sigma}_S^2$ to the test-set variance $\text{Var}_{h\sim\mathcal{H}_\mathcal{A}}(\text{err}_S(h))$ across a range of training durations. When training for 4 epochs, we estimate that the standard deviation of true error is $\sqrt{\hat{\sigma}_S^2} = 0.52\%$, indicating significant differences in model quality between trainings of this duration. In contrast, when training for 64 epochs, we estimate that the true standard deviation is only 0.033%. In comparison, that configuration's test-set standard deviation is 0.149%. We obtain a similarly small estimate for ImageNet trainings in Section C. These findings indicate that when training to convergence, there is little variation in model quality (*i.e.*, expected performance with respect to new batches of data from the test-distribution) between runs of training.

Having confirmed that true distribution-wise variance is small, it still remains to explain why there is high variance on the finite test-set in the first place. We investigate this in the next section.

### 3.4 CALIBRATION IMPLIES (FINITE-SAMPLE) VARIATION

In this section we prove that variance in test-set accuracy between runs of training is an inevitable consequence of the observation that ensembles of trained networks are well-calibrated (Jiang et al., 2021). Our analysis leads us to a simple formula which accurately estimates variance for binary classification problems.

Classical machine learning algorithms based on convex loss functions can have neither test-set nor test-distribution variance, because training always converges to a single global optimum. As an example, we show in Figure 7 (left) that repeated runs of training of a regularized linear model on CIFAR-10 leads to below 0.01% variance in test-set accuracy. On the other hand, neural networks have many optima (Auer et al., 1995; Choromanska et al., 2015), so that every run of training can potentially lead to a different solution. Despite this theoretical property, in the previous section we showed that the variance in true error between runs of training is in fact quite small. What then explains the significant variance in test-set error?

We show that the following property, which Jiang et al. (2021) demonstrated approximately holds for neural network trainings in practice, is connected to their high variance on test-sets.

**Definition 2.** *The training algorithm $\mathcal{A}$ satisfies **class-wise calibration** (Jiang et al., 2021) if for every class $y \in \mathcal{Y}$ and confidence level $q \in [0, 1]$,*

$$P_{(x,y)\sim\mathcal{D}}\big(y = c \mid P_{h\sim\mathcal{H}_\mathcal{A}}(h(x) = c) = q\big) = q. \tag{4}$$

As an explanatory example, if we let $S' \subset S$ be the subset of test images which are classified by 30-40% of independently trained neural networks as "cat," then 30-40% of $S'$ really will be cats.

**Theorem 3.** *Given a stochastic learning algorithm $\mathcal{A}$ for binary classification, if it is class-wise calibrated, then its expected variance on a test-set of size $n$ is*

$$\mathbb{E}_{S\sim\mathcal{D}^n}\left[\operatorname*{Var}_{h\sim\mathcal{H}_\mathcal{A}}(\mathrm{err}_S(h))\right] = \frac{\mathrm{err}(\mathcal{A})}{2n} + (1-\tfrac{1}{n})\cdot \operatorname*{Var}_{h\sim\mathcal{H}_\mathcal{A}}(\mathrm{err}(h)) \tag{5}$$

Proof is provided in Section B.3. In the previous section we showed that the true variance $\mathrm{Var}_h(\mathrm{err}(h))$ is small in practice. Therefore, the above theorem practically reduces to the following simple formula:

$$\mathbb{E}_S[\mathrm{Var}_h(\mathrm{err}_S(h)] \approx \mathrm{err}(\mathcal{A})/2n \tag{6}$$

In Figure 6 we use this to predict the variance in test-set accuracy across 511 different binary classification tasks derived from CIFAR-10, where we substitute the average test-set errors as a cheap approximation to the true error $\mathrm{err}(\mathcal{A})$. It is a good empirical fit, with $R^2 = 0.996$ across the collection of tasks. We also compare to the commonly used binomial assumption (Dietterich, 1998; Raschka, 2018), which yields a larger variance estimate of $\mathrm{err}(1 - \mathrm{err})/n$. The formula based on class-wise calibration gives variance estimates which are $70.5\times$ more accurate than those based on the binomial assumption, in terms of their average squared distance to the empirical variance.

We additionally provide the following lower bound for general multiclass classification.

**Theorem 4.** *Given a learning algorithm $\mathcal{A}$ for $k$-way classification, if it is class-wise calibrated, then its expected variance on a test-set of size $n$ is*

$$\mathbb{E}_{S\sim\mathcal{D}^n}\left[\operatorname*{Var}_{h\sim\mathcal{H}_\mathcal{A}}(\mathrm{err}_S(h))\right] \geq \frac{\mathrm{err}(\mathcal{A})}{nk} \tag{7}$$

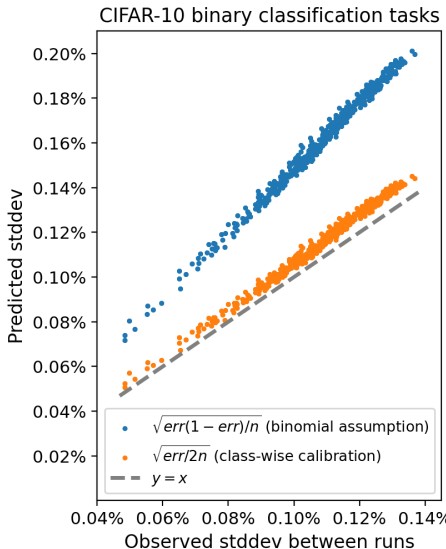

Figure 6: **Predicting test-set variance.** Across hundreds of tasks, the formula given by class-wise calibration accurately predicts the standard deviation of test-set error. In contrast, the formula given by the binomial assumption is inaccurate.

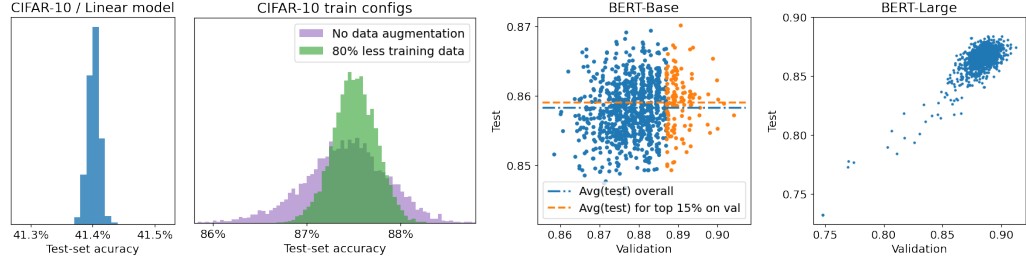

Figure 7: (Far left:) Training a regularized linear model has very little variance between runs. (Center left:) Removing either data augmentation, or 80% of training data from CIFAR-10 training, reduces the average accuracy to around 87.5%. But the former produces much more variance between runs than the latter. (Right two:) When finetuning BERT_BASE on MRPC, variations between runs in terms of performance on the validation and test sets are not strongly correlated. On the other hand, BERT_LARGE has significant true instability.

Proof is provided in Section B.4. Although it is still largely a mystery why neural network trainings satisfy class-wise calibration, together these theorems show that their variance on finite test-sets is an inevitable consequence of that property.

## 4 ADDITIONAL EXPERIMENTS

In this section we apply the tools developed in the preceding sections to BERT finetuning, and conduct preliminary investigations of the effect on data augmentation, learning rate, and distribution-shift on variance. We additionally include a replication study of Summers & Dinneen (2021) in Section D, which confirms their findings that variance is caused by extreme sensitivity to initial conditions rather than any particular stochastic factor like the initialization or data ordering.

### 4.1 BERT FINETUNING

In this section we study BERT (Devlin et al., 2018) finetuning, where previous works have reported significant variance between runs (Devlin et al., 2018; Dodge et al., 2020; Mosbach et al., 2020). Our contribution is to use the tools we developed in Section 3 to clearly differentiate the behavior of BERT_LARGE from BERT_BASE, arguing that although both models exhibit high variance in their test-set error rates, only the former has high variance in its true error rate.

For our experiment, we finetune pretrained checkpoints of both models 1,000 times each on the MRPC (Dolan & Brockett, 2005) task. The test-set error rate of BERT_BASE has a standard deviation of 0.80% between runs of finetuning, and BERT_LARGE has a stddev of 2.24%. In Figure 7 (right) we show that for BERT_BASE, the validation and test splits of MRPC are close to decorrelated in terms of finetuned model performance, recalling Section 3.1. In particular, the top 15% of seeds in terms of validation-set performance achieve only 0.09% higher performance than average on the test-set. This observation is reinforced by the use of Equation 3, which computes a distribution-wise stddev of only 0.21% for BERT_BASE, compared to 2.08% for BERT_LARGE, where there is a clear correlation. We therefore conclude that, despite both models appearing to have high test-set variance, in fact only BERT_LARGE has substantial variance in its true error rate. Our contribution ends at making this distinction; we do not speculate as to the reasons underlying this instability.

### 4.2 THE EFFECT OF DATA AUGMENTATION

In this section we look at the effect of data augmentation on variance. In Figure 7 (center left) we compare two ablations: first, removing a fixed 80% of training data, and second, removing data augmentation. While both configurations achieve a similar mean accuracy of 87.5%, the augmentation-free training has over $3.5\times$ more variance between runs. We also observe that the ensemble accuracy of the augmentation-free networks is higher, reaching 91.2%, compared to the reduced-data ensemble, which achieves only 89.8%. Based on these observations, we speculate that one role of data augmentation may be to reduce variance between runs.

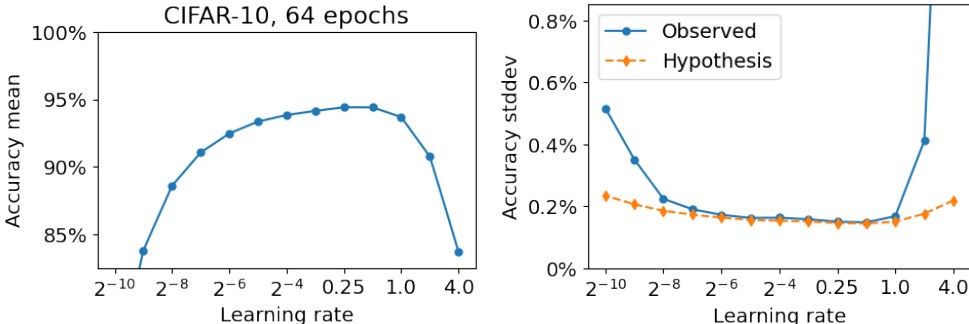

Figure 8: **Accuracy is maximized by the largest learning rate without excess variance.** Across learning rates, we plot the observed stddev of test-set accuracy and that predicted by the independent errors assumption. We observe that the best learning rate is the largest one which does not induce significant excess variance.

### 4.3 THE EFFECT OF LEARNING RATE

In this section we investigate the relationship between learning rate and variance. Our experiment is to execute 1,000 64-epoch CIFAR-10 trainings for each binary-power learning rate between $2^{-10}$ and $2^2$. For each setting, we measure the mean and variance of test-set accuracy. We observe that the learning rate 0.5 yields both the highest mean and the lowest variance. Raising it to 1.0 causes the standard deviation of test-set accuracy to increase from 0.148% to 0.168%. This may seem insignificant, but Equation 3 estimates that it coincides with a $5\times$ increase in distribution-wise variance. We therefore conjecture that, as a general property of neural network trainings, the optimal learning rate is the largest one which does not induce significant distribution-wise variance.

### 4.4 THE EFFECT OF DISTRIBUTION SHIFT

In this subsection we summarize our results on distribution shift which are fully described in Section C. Our experimental setup is to train 1,000 ResNets on ImageNet (with identical hyperparameters), and then use Equation 3 to estimate the distribution-wise variance on the validation set, ImageNet-V2 (Recht et al., 2019), and three distribution-shifted sets. We find that the main validation set and ImageNet-V2 have standard deviation at or below 0.07%, while the three distribution-shifted sets all have at least $6\times$ more variance. That is, distribution-wise variance is large for precisely those test-sets which have a shifted distribution relative to the training set.

## 5 DISCUSSION

A central focus of paper was the distinction between a model's observed error rate on a test-set, and its true error rate on the test-distribution. We showed that although the true error rate is not directly accessible (since we can't sample an infinite number of test examples), it is nevertheless possible to acquire an unbiased estimate of its variance via Equation 3. And we found that for standard trainings, this quantity is quite small, with the standard deviation of true error being only around 0.03% for both CIFAR-10 and ImageNet. Our takeaway is that for standard trainings, *even though some random seeds lead to higher or lower performance on the test-set, they are all nearly equal on the test-distribution*.

However, we found two exceptions to this takeaway. The first fairly obvious exception is trainings which have pathological instability, such as BERT$_{\text{LARGE}}$ finetuning where the accuracy can vary by more than 15% (Section 4.1, Dodge et al. (2020)). The second more important exception is trainings whose test distribution is shifted relative to the training distribution, for which case we observed many times more variance (Section C). Understanding why variance appears alongside distribution shift is a task whose solution we look forward to in future work.

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

# A    TRAINING HYPERPARAMETERS

## A.1    CIFAR-10

For our core experiments we train a ResNet on CIFAR-10. Our network architecture is the same as was used in Ilyas et al. (2022); namely, a 9-layer ResNet originally derived from Page (2019). Our 0-epoch configuration corresponds to a randomly initialized network. Our 4, 16, and 64-epoch configurations all train using SGD with a learning rate of 0.5, a momentum of 0.9, and a weight decay of 5e-4, with the learning rate linearly ramped down to zero by the end of training. We train using random flipping, 2-pixel translation, and 12-pixel Cutout (DeVries & Taylor, 2017) data augmentations. We use a batch size 500 and load data using the FFCV (Leclerc et al., 2022) library. Our training script is publicly available at `https://github.com/KellerJordan/ffcv-cifar/blob/master/train.py`. The 64-epoch configuration attains an average accuracy of 94.42% without any test-time augmentation. We execute each configuration 60,000 times, generating 240,000 sets of test-set predictions, which form our object of study for Section 3.

## A.2    IMAGENET

For our ImageNet experiments we train ResNet-18s using standard random flip and random resized crop data augmentations. We train at resolution 192 for 100 epochs with batch size 1024, using SGD-momentum with learning rate 0.5, momentum 0.9, and weight decay 5e-5. We linearly ramp the learning rate up from 5e-5 to 0.5 by epoch 2, and then down to zero by the end of training. We use the FFCV (Leclerc et al., 2022) dataloader here as well.

## A.3    BERT FINETUNING

We finetune $\text{BERT}_{\text{BASE}}$ and $\text{BERT}_{\text{LARGE}}$ on the MRPC dataset. We train for 3 epochs at batch size 16, using Adam (Kingma & Ba, 2014) with default hyperparameters other than the learning rate, which is linearly annealed from a maximum of 2e-5 down to zero by the end of training.

# B    PROOFS

We first prove the following two lemmas, using the notation of Section 2.

**Lemma 1.** *The variance of the overall error rate is equal to the expected covariance between error rates on a pair of examples.*

$$\operatorname*{Var}_{h\sim\mathcal{H}_{\mathcal{A}}}\left(\text{err}(h)\right) = \mathop{\mathbb{E}}_{(x_1,y_1),(x_2,y_2)\sim\mathcal{D}^2}\left[\operatorname*{Cov}_{h\sim\mathcal{H}_{\mathcal{A}}}\left(\text{err}_{x_1,y_1}(h), \text{err}_{x_2,y_2}(h)\right)\right]$$

*Proof.*

$$\text{Var}_{h \sim \mathcal{H}_\mathcal{A}} (\text{err}(h)) = \mathbb{E}_{h \sim \mathcal{H}_\mathcal{A}} \left[ \left( \text{err}(h) - \mathbb{E}_h[\text{err}(h)] \right)^2 \right]$$

$$= \mathbb{E}_h \left[ \left( \mathbb{E}_{x,y} \left[ \text{err}_{x,y}(h) \right] - \mathbb{E}_h \left[ \mathbb{E}_{x,y} \left[ \text{err}_{x,y}(h) \right] \right] \right)^2 \right]$$

$$= \mathbb{E}_h \left[ \left( \mathbb{E}_{x,y} \left[ \text{err}_{x,y}(h) \right] - \mathbb{E}_{x,y} \left[ \mathbb{E}_h [\text{err}_{x,y}(h)] \right] \right)^2 \right]$$

$$= \mathbb{E}_h \left[ \left( \mathbb{E}_{x,y} \left[ \text{err}_{x,y}(h) - \mathbb{E}_h[\text{err}_{x,y}(h)] \right] \right)^2 \right]$$

$$= \mathbb{E}_h \left[ \mathbb{E}_{x_1,y_1} \left[ \text{err}_{x_1,y_1}(h) - \mathbb{E}_h[\text{err}_{x_1,y_1}(h)] \right] \cdot \mathbb{E}_{x_2,y_2} \left[ \text{err}_{x_2,y_2}(h) - \mathbb{E}_h[\text{err}_{x_2,y_2}(h)] \right] \right]$$

$$= \mathbb{E}_h \mathbb{E}_{x_1,y_1} \mathbb{E}_{x_2,y_2} \left[ (\text{err}_{x,y}(h) - \mathbb{E}_h[\text{err}_{x,y}(h)])(\text{err}_{x_2,y_2}(h) - \mathbb{E}_h[\text{err}_{x_2,y_2}(h)]) \right]$$

$$= \mathbb{E}_{x_1,y_1} \mathbb{E}_{x_2,y_2} \mathbb{E}_h \left[ (\text{err}_{x_1,y_1}(h) - \mathbb{E}_h[\text{err}_{x_1,y_1}(h)])(\text{err}_{x_2,y_2}(h) - \mathbb{E}_h[\text{err}_{x_2,y_2}(h)]) \right]$$

$$= \mathbb{E}_{(x_1,y_1),(x_2,y_2) \sim \mathcal{D}^2} \left[ \text{Cov}_{h \sim \mathcal{H}_\mathcal{A}} (\text{err}_{x_1,y_1}(h), \text{err}_{x_2,y_2}(h)) \right].$$

$\square$

**Lemma 2.** *For an IID test-set $S = ((x_1, y_1), \ldots, (x_n, y_n))$, the expected variance in the test error rate can be decomposed into a mixture of distribution-wise and example-wise variances.*

$$\mathbb{E}_{S \sim \mathcal{D}^n} \left[ \text{Var}_{h \sim \mathcal{H}_\mathcal{A}} (\text{err}_S(h)) \right] = (1 - 1/n) \cdot \text{Var}_{h \sim \mathcal{H}_\mathcal{A}} (\text{err}(h)) + (1/n) \cdot \mathbb{E}_{(x,y) \sim \mathcal{D}} \left[ \text{Var}_{h \sim \mathcal{H}_\mathcal{A}} (\text{err}_{x,y}(h)) \right]$$

*Proof.*

$$\mathbb{E}_{S \sim \mathcal{D}^n} \left[ \text{Var}_{h \sim \mathcal{H}_\mathcal{A}} (\text{err}_S(h)) \right] = \mathbb{E}_S \left[ \text{Var}_h \left( \frac{1}{n} \sum_{i=1}^n \text{err}_{x_i,y_i}(h) \right) \right]$$

$$= \mathbb{E}_S \left[ \frac{1}{n^2} \sum_{i=1}^n \sum_{j=1}^n \text{Cov}_h(\text{err}_{x_i,y_i}(h), \text{err}_{x_j,y_j}(h)) \right]$$

$$= \frac{1}{n^2} \sum_{i=1}^n \sum_{j=1}^n \mathbb{E}_S \left[ \text{Cov}_h(\text{err}_{x_i,y_i}(h), \text{err}_{x_j,y_j}(h)) \right]$$

$$= \frac{1}{n^2} \sum_{i=1}^n \mathbb{E}_S \left[ \text{Var}_h(\text{err}_{x_i,y_i}(h)) \right] + \frac{1}{n^2} \sum_{i=1}^n \sum_{j \neq i} \mathbb{E}_S \left[ \text{Cov}_h(\text{err}_{x_i,y_i}(h), \text{err}_{x_j,y_j}(h)) \right]$$

$$= \frac{1}{n^2} \sum_{i=1}^n \mathbb{E}_{x_i,y_i} \left[ \text{Var}_h(\text{err}_{x_i,y_i}(h)) \right] + \frac{1}{n^2} \sum_{i=1}^n \sum_{j \neq i} \mathbb{E}_{(x_i,y_i),(x_j,y_j)} \left[ \text{Cov}_h(\text{err}_{x_i,y_i}(h), \text{err}_{x_j,y_j}(h)) \right]$$

$$= \frac{n}{n^2} \mathbb{E}_{x,y} \left[ \text{Var}_h(\text{err}_{x,y}(h)) \right] + \frac{n(n-1)}{n^2} \mathbb{E}_{(x_1,y_1),(x_2,y_2)} \left[ \text{Cov}_h(\text{err}_{x_1,y_1}(h), \text{err}_{x_2,y_2}(h)) \right]$$

$$= (1/n) \cdot \mathbb{E}_{(x,y) \sim \mathcal{D}} \left[ \text{Var}_{h \sim \mathcal{H}_\mathcal{A}} (\text{err}_{x,y}(h)) \right] + (1 - 1/n) \cdot \text{Var}_{h \sim \mathcal{H}_\mathcal{A}} (\text{err}(h))$$

Where the last step uses Lemma 1.

$\square$

## B.1 Theorem 1

**Theorem 1.** *In expectation, variance in test-set accuracy overestimates variance in true error.*

$$\mathbb{E}_{S \sim \mathcal{D}^n} \left[ \operatorname*{Var}_{h \sim \mathcal{H}_{\mathcal{A}}} (\operatorname{err}_S(h)) \right] \geq \operatorname*{Var}_{h \sim \mathcal{H}_{\mathcal{A}}} (\operatorname{err}(h))$$

*Proof.* The difference between the two terms is

$$\mathbb{E}_{S \sim \mathcal{D}^n} \left[ \operatorname*{Var}_{h \sim \mathcal{H}_{\mathcal{A}}} (\operatorname{err}_S(h)) \right] - \operatorname*{Var}_{h \sim \mathcal{H}_{\mathcal{A}}} (\operatorname{err}(h))$$

$$= \frac{1}{n} \left( \mathbb{E}_{(x,y) \sim \mathcal{D}} \left[ \operatorname*{Var}_{h \sim \mathcal{H}_{\mathcal{A}}} (\operatorname{err}_{x,y}(h)) \right] - \operatorname*{Var}_{h \sim \mathcal{H}_{\mathcal{A}}} (\operatorname{err}(h)) \right)$$

$$= \frac{1}{n} \left( \mathbb{E}_{x,y} \left[ \operatorname*{Var}_{h}(\operatorname{err}_{x,y}(h)) \right] - \mathbb{E}_{(x_1,y_1),(x_2,y_2)} \left[ \operatorname*{Cov}_{h}(\operatorname{err}_{x_1,y_1}(h), \operatorname{err}_{x_2,y_2}(h)) \right] \right)$$

$$= \frac{1}{n} \left( 0.5 \cdot \mathbb{E}_{x_1,y_1} \left[ \operatorname*{Var}_{h}(\operatorname{err}_{x_1,y_1}(h)) \right] + 0.5 \cdot \mathbb{E}_{x_2,y_2} \left[ \operatorname*{Var}_{h}(\operatorname{err}_{x_2,y_2}(h)) \right] \right.$$

$$\left. - \mathbb{E}_{(x_1,y_1),(x_2,y_2)} \left[ \operatorname*{Cov}_{h}(\operatorname{err}_{x_1,y_1}(h), \operatorname{err}_{x_2,y_2}(h)) \right] \right)$$

$$= \frac{1}{2n} \mathbb{E}_{(x_1,y_1),(x_2,y_2)} \left[ \operatorname*{Var}_{h}(\operatorname{err}_{x_1,y_1}(h)) + \operatorname*{Var}_{h}(\operatorname{err}_{x_2,y_2}(h)) - 2 \operatorname*{Cov}_{h}(\operatorname{err}_{x_1,y_1}(h), \operatorname{err}_{x_2,y_2}(h)) \right]$$

$$\geq \frac{1}{2n} \mathbb{E}_{(x_1,y_1),(x_2,y_2)} \left[ \operatorname*{Var}_{h}(\operatorname{err}_{x_1,y_1}(h)) + \operatorname*{Var}_{h}(\operatorname{err}_{x_2,y_2}(h)) - 2 \sqrt{\operatorname*{Var}_{h}(\operatorname{err}_{x_1,y_1}(h)) \operatorname*{Var}_{h}(\operatorname{err}_{x_2,y_2}(h))} \right]$$

$$= \frac{1}{2n} \mathbb{E}_{(x_1,y_1),(x_2,y_2)} \left[ \left( \sqrt{\operatorname*{Var}_{h}(\operatorname{err}_{x_1,y_1}(h))} - \sqrt{\operatorname*{Var}_{h}(\operatorname{err}_{x_2,y_2}(h))} \right)^2 \right]$$

$$\geq 0.$$

where the first two steps use Lemma 1 and then Lemma 2. $\qquad \square$

## B.2 Theorem 2

**Theorem 2.** *The following quantity is an unbiased estimator for true variance* $\operatorname{Var}_{h \sim \mathcal{H}_{\mathcal{A}}}(\operatorname{err}(h))$.

$$\hat{\sigma}_S^2 = \frac{n}{n-1} \left( \operatorname*{Var}_{h \sim \mathcal{H}_{\mathcal{A}}} (\operatorname{err}_S(h)) - \frac{1}{n^2} \sum_{i=1}^{n} \operatorname*{Var}_{h \sim \mathcal{H}_{\mathcal{A}}} (\operatorname{err}_{x_i,y_i}(h)) \right)$$

*Proof.*

$$\operatorname*{Var}_{h \sim \mathcal{H}_{\mathcal{A}}} (\operatorname{err}(h)) = \frac{n}{n-1} \left( \mathbb{E}_{S \sim \mathcal{D}^n} \left[ \operatorname*{Var}_{h \sim \mathcal{H}_{\mathcal{A}}} (\operatorname{err}_S(h)) \right] - (1/n) \cdot \mathbb{E}_{(x,y) \sim \mathcal{D}} \left[ \operatorname*{Var}_{h \sim \mathcal{H}_{\mathcal{A}}} (\operatorname{err}_{x,y}(h)) \right] \right)$$

$$= \frac{n}{n-1} \left( \mathbb{E}_{S \sim \mathcal{D}^n} \left[ \operatorname*{Var}_{h \sim \mathcal{H}_{\mathcal{A}}} (\operatorname{err}_S(h)) \right] - (1/n) \cdot \mathbb{E}_{S \sim \mathcal{D}^n} \left[ \frac{1}{n} \sum_{i=1}^{n} \operatorname*{Var}_{h \sim \mathcal{H}_{\mathcal{A}}} (\operatorname{err}_{x_i,y_i}(h)) \right] \right)$$

$$= \mathbb{E}_{S \sim \mathcal{D}^n} \left[ \frac{n}{n-1} \left( \operatorname*{Var}_{h \sim \mathcal{H}_{\mathcal{A}}} (\operatorname{err}_S(h)) - \frac{1}{n^2} \sum_{i=1}^{n} \operatorname*{Var}_{h \sim \mathcal{H}_{\mathcal{A}}} (\operatorname{err}_{x_i,y_i}(h)) \right) \right]$$

Where the first equality is a rearrangement of Lemma 2. $\qquad \square$

The quantity $\hat{\sigma}_S^2$ is also equal to $\binom{n}{2}^{-1} \sum_{i=1}^{n} \sum_{j \neq i} \operatorname{Cov}_{h \sim \mathcal{H}_{\mathcal{A}}}(\operatorname{err}_{x_i,y_i}(h), \operatorname{err}_{x_j,y_j}(h))$. Comparing this formula to Lemma 1 may help provide intuition for why it is an estimator for the true variance. The formulation given in Theorem 2 looks less intuitive, but the benefit is that we only have to calculate $n$ separate variances, rather than $\binom{n}{2}$ separate covariance values.

We note that the proofs of Theorem 1 and Theorem 2 do not assume anything about the error function $\text{err}_{x,y}(h)^2$, so, *e.g.*, they are also true for regression tasks.

## B.3 THEOREM 3

**Theorem 3.** *Let $\mathcal{A}$ be a training algorithm for binary classification which satisfies class-wise calibration (Definition 2). Then the expected variance on an IID test-set of size $n$ is*

$$\underset{S \sim \mathcal{D}^n}{\mathbb{E}} \left[ \underset{h \sim \mathcal{H}_{\mathcal{A}}}{\text{Var}} (\text{err}_S(h)) \right] = \frac{\text{err}(\mathcal{A})}{2n} + (1 - 1/n) \cdot \underset{h \sim \mathcal{H}_{\mathcal{A}}}{\text{Var}} (\text{err}(h))$$

*Proof.* Define the random variable $q(x) = \mathbb{E}_{h \sim \mathcal{H}_{\mathcal{A}}}[1\{h(x) = 1\}]$ to be the proportion of training runs which classify $x$ as positive. We first compute a formula for $\text{err}(\mathcal{A})$ in terms of $q$. By the usual laws of conditional expectation we have

$$
\begin{aligned}
\text{err}(\mathcal{A}) &= \underset{x,y,h}{\mathbb{E}} [\text{err}_{x,y}(h)] = \underset{q}{\mathbb{E}}[\underset{x,y,h}{\mathbb{E}} [\text{err}_{x,y}(h) \mid q]] \\
&= \underset{q}{\mathbb{E}}[\underset{x,y,h}{\mathbb{E}} [1\{h(x) \neq y\} \mid q]] \\
&= \underset{q}{\mathbb{E}}[\underset{x,y,h}{\mathbb{E}} [1\{y = 0\}1\{h(x) = 1\} + 1\{y = 1\}1\{h(x) = 0\} \mid q]] \\
&= \underset{q}{\mathbb{E}}[\underset{x,y}{\mathbb{E}} [\underset{h}{\mathbb{E}}[1\{y = 0\}1\{h(x) = 1\} + 1\{y = 1\}1\{h(x) = 0\} \mid q] \mid q]] \\
&= \underset{q}{\mathbb{E}}[\underset{x,y}{\mathbb{E}} [1\{y = 0\} \underset{h}{\mathbb{E}}[1\{h(x) = 1\} \mid q] + 1\{y = 1\} \underset{h}{\mathbb{E}}[1\{h(x) = 0\} \mid q] \mid q]] \\
&= \underset{q}{\mathbb{E}}[\underset{x,y}{\mathbb{E}} [q \cdot 1\{y = 0\} + (1 - q) \cdot 1\{y = 1\} \mid q]] \\
&= \underset{q}{\mathbb{E}}[q(1 - E_{x,y}[1\{y = 1\} \mid q]) + (1 - q) \underset{x,y}{\mathbb{E}} [1\{y = 1\} \mid q]] \\
&= \underset{q}{\mathbb{E}}[q(1 - \underset{x,y}{\mathbb{E}} [1\{y = 1\} \mid q]) + (1 - q)(\underset{x,y}{\mathbb{E}} [1\{y = 1\} \mid q])].
\end{aligned}
$$

Using the assumption of class-wise calibration, this is equal to

$$\underset{q}{\mathbb{E}}[q(1 - q) + (1 - q)q] = \underset{q}{\mathbb{E}}[2q(1 - q)].$$

Next we analyze the example-wise variance. We have

$$
\begin{aligned}
\underset{x,y}{\mathbb{E}} [\underset{h}{\text{Var}}(\text{err}_{x,y}(h))] &= \underset{q}{\mathbb{E}}[\underset{x,y}{\mathbb{E}} [\underset{h}{\text{Var}}(1\{h(x) \neq y)\}) \mid q(x) = q]] \\
&= \underset{q}{\mathbb{E}}[\underset{x,y}{\mathbb{E}} [\underset{h}{\mathbb{E}}[1\{h(x) \neq y\}](1 - \underset{h}{\mathbb{E}}[1\{h(x) \neq y\}]) \mid q]] \\
&= \underset{q}{\mathbb{E}}[\underset{x,y}{\mathbb{E}} [\underset{h}{\mathbb{E}}[1\{h(x) = 1\}] \underset{h}{\mathbb{E}}[1\{h(x) = 0\}] \mid q]] \\
&= \underset{q}{\mathbb{E}}[\underset{x,y}{\mathbb{E}} [q(1 - q) \mid q]] \\
&= \underset{q}{\mathbb{E}}[q(1 - q)].
\end{aligned}
$$

The second equality uses the formula for variance of a Bernoulli variable. The third equality uses the fact that, regardless of whether $y = 0$ or $y = 1$, the product $\mathbb{E}_h[1\{h(x) \neq y\}](1 - \mathbb{E}_h[1\{h(x) \neq y\}])$ is equal to $1\{h(x) = 1\}] \mathbb{E}_h[1\{h(x) = 0\}]$. The fourth equality applies the assumption of class-wise calibration.

Finally using Lemma 2 and the previous two results we get

$$
\begin{aligned}
\underset{S \sim \mathcal{D}^n}{\mathbb{E}} \left[ \underset{h \sim \mathcal{H}_{\mathcal{A}}}{\text{Var}} (\text{err}_S(h)) \right] &= (1 - 1/n) \cdot \underset{h \sim \mathcal{H}_{\mathcal{A}}}{\text{Var}} (\text{err}(h)) + (1/n) \cdot \underset{(x,y) \sim \mathcal{D}}{\mathbb{E}} \left[ \underset{h \sim \mathcal{H}_{\mathcal{A}}}{\text{Var}} (\text{err}_{x,y}(h)) \right] \\
&= (1 - 1/n) \cdot \underset{h \sim \mathcal{H}_{\mathcal{A}}}{\text{Var}} (\text{err}(h)) + (1/n) \cdot \underset{q}{\mathbb{E}}[q(1 - q)] \\
&= \frac{\text{err}(\mathcal{A})}{2n} + (1 - 1/n) \cdot \underset{h \sim \mathcal{H}_{\mathcal{A}}}{\text{Var}} (\text{err}(h))
\end{aligned}
$$

$\square$

---

[2]That is, other than it being non-pathological enough to allow the interchanges of expectation via Fubini's theorem, *e.g.*, nonnegative suffices.

## B.4 THEOREM 4

**Theorem 4.** *Let $\mathcal{A}$ be a training algorithm for $k$-way classification which satisfies class-wise calibration (Definition 2). Then the expected variance on an IID test-set of size $n$ is at least*

$$\mathop{\mathbb{E}}_{S \sim \mathcal{D}^n} \left[ \mathop{\mathrm{Var}}_{h \sim \mathcal{H}_\mathcal{A}} (\mathrm{err}_S(h)) \right] \geq \frac{\mathrm{err}(\mathcal{A})}{nk}$$

*Proof.* For each class $c \in \{1, \ldots, k\}$, define the random variable $q_c(x) = \mathbb{E}_{h \sim \mathcal{H}_\mathcal{A}}[1\{h(x) = c\}]$ to be the proportion of runs of training which classify $x$ as $c$. Let $q(x) = (q_1(x), \ldots, q_k(x))$ be the vector of these variables. The laws of conditional expectation yield the following expression for the expected error.

$$
\begin{aligned}
\mathrm{err}(\mathcal{A}) &= \mathop{\mathbb{E}}_{x,y,h} [\mathrm{err}_{x,y}(h)] \\
&= \mathop{\mathbb{E}}_{q} [ \mathop{\mathbb{E}}_{x,y,h} [\mathrm{err}_{x,y}(h) \mid q(x) = q]] \\
&= \mathop{\mathbb{E}}_{q} [ \mathop{\mathbb{E}}_{x,y,h} [1\{h(x) \neq y\} \mid q]] \\
&= \mathop{\mathbb{E}}_{q} \left[ \mathop{\mathbb{E}}_{x,y,h} \left[ \sum_{c=1}^{k} 1\{y = c\} 1\{h(x) \neq c\} \mid q \right] \right] \\
&= \mathop{\mathbb{E}}_{q} \left[ \mathop{\mathbb{E}}_{x,y} \left[ \mathop{\mathbb{E}}_{h} \left[ \sum_{c=1}^{k} 1\{y = c\} 1\{h(x) \neq c\} \mid q \right] \mid q \right] \right] \\
&= \mathop{\mathbb{E}}_{q} \left[ \sum_{c=1}^{k} (1 - q_c) \mathop{\mathbb{E}}_{x,y} [1\{y = c\} \mid q] \right] \\
&= \mathop{\mathbb{E}}_{q} \left[ \sum_{c=1}^{k} q_c (1 - q_c) \right].
\end{aligned}
$$

The last step uses the assumption of class-wise calibration. We next derive a related expression for the example-wise variance.

$$
\begin{aligned}
\mathop{\mathbb{E}}_{x,y} [\mathop{\mathrm{Var}}_{h}(\mathrm{err}_{x,y}(h))] &= \mathop{\mathbb{E}}_{q} [ \mathop{\mathbb{E}}_{x,y} [\mathop{\mathrm{Var}}_{h} (1\{h(x) \neq y\}) ) \mid q(x) = q]] \\
&= \mathop{\mathbb{E}}_{q} [ \mathop{\mathbb{E}}_{x,y} [\mathop{\mathbb{E}}_{h}[1\{h(x) \neq y\} \mid q](1 - \mathop{\mathbb{E}}_{h}[1\{h(x) \neq y\} \mid q]) \mid q]] \\
&= \mathop{\mathbb{E}}_{q} [ \mathop{\mathbb{E}}_{x,y} [q_y(1 - q_y) \mid q]] \\
&= \mathop{\mathbb{E}}_{q} \left[ \mathop{\mathbb{E}}_{x,y} \left[ \sum_{c=1}^{k} 1\{y = c\} q_c (1 - q_c) \mid q \right] \right] \\
&= \mathop{\mathbb{E}}_{q} \left[ \sum_{c=1}^{k} \mathop{\mathbb{E}}_{x,y} [1\{y = c\} \mid q] \cdot q_c (1 - q_c) \right] \\
&= \mathop{\mathbb{E}}_{q} \left[ \sum_{c=1}^{k} q_c^2 (1 - q_c) \right]
\end{aligned}
$$

We now analyze the ratio between $\sum_{c=1}^{k} q_c^2(1 - q_c)$ and $\sum_{c=1}^{k} q_c(1 - q_c)$.

Without loss of generality, let $q_1 \leq q_2 \leq \cdots \leq q_k$ be in nondecreasing order. Then we have

$$
\begin{aligned}
\sum_{c=1}^{k} q_c^2(1 - q_c) &= k \cdot \left( \frac{1}{k} \sum_{c=1}^{k} q_c \cdot q_c(1 - q_c) \right) \\
&\geq k \cdot \left( \frac{1}{k} \sum_{c=1}^{k} q_c \right) \left( \frac{1}{k} \sum_{c=1}^{k} q_c(1 - q_c) \right) \\
&= \frac{1}{k} \sum_{c=1}^{k} q_c(1 - q_c).
\end{aligned}
$$

The inequality step is due to an application of Chevychev's sum inequality (Hardy et al., 1934), which is possible because the series $q_1(1 - q_1), \ldots, q_k(1 - q_k)$ is nondecreasing, which we prove as follows.

We first recall that $\sum_{c=1}^{k} q_c = 1$, and that we assumed without loss of generality that $q_1 \leq \cdots \leq q_k$. For the first $k - 1$ terms, the monotonicity of the mapping $x \mapsto x(1 - x)$ on the interval $[0, 1/2]$, combined with the fact that $q_c \leq 1/2$ for $c \in \{1, \ldots, k - 1\}$, implies $q_1(1 - q_1) \leq \cdots \leq q_{k-1}(1 - q_{k-1})$. It remains to show that $q_{k-1}(1 - q_{k-1}) \leq q_k(1 - q_k)$. If $q_k \leq 1/2$, then this is again due to the monotonicity of $x \mapsto x(1-x)$ on $[0, 1/2]$. Otherwise if $q_k \geq 1/2$, then combining $q_{k-1} \leq 1 - q_k$ and $(1 - q_k) \leq 1/2$ yields $q_{k-1}(1 - q_{k-1}) \leq (1 - q_k)(1 - (1 - q_k)) = q_k(1 - q_k)$. Either way, we have shown that $q_1(1 - q_1) \leq \cdots \leq q_k(1 - q_k)$ is in nondecreasing order, allowing the application of Chevychev's sum inequality above.

Putting Lemma 2 together with the above results, as follows, yields the theorem.

$$
\begin{aligned}
\mathop{\mathbb{E}}_{S \sim \mathcal{D}^n} \left[ \mathop{\text{Var}}_{h \sim \mathcal{H}_{\mathcal{A}}} (\text{err}_S(h)) \right] &= (1 - 1/n) \cdot \mathop{\text{Var}}_{h \sim \mathcal{H}_{\mathcal{A}}} (\text{err}(h)) + (1/n) \cdot \mathop{\mathbb{E}}_{(x,y) \sim \mathcal{D}} \left[ \mathop{\text{Var}}_{h \sim \mathcal{H}_{\mathcal{A}}} (\text{err}_{x,y}(h)) \right] \\
&\geq (1/n) \cdot \mathop{\mathbb{E}}_{x,y} \left[ \mathop{\text{Var}}_{h} (\text{err}_{x,y}(h)) \right] \\
&= \frac{1}{n} \mathop{\mathbb{E}}_{q} \left[ \sum_{c=1}^{k} q_c^2(1 - q_c) \right] \\
&\geq \frac{1}{nk} \mathop{\mathbb{E}}_{q} \left[ \sum_{c=1}^{k} q_c(1 - q_c) \right] \\
&= \frac{\text{err}(\mathcal{A})}{nk}.
\end{aligned}
$$

$\square$

## B.5 REPLICATION OF THE MAIN RESULT FROM JIANG ET AL. (2021)

Because it is theoretically related to our results, we include a simplified proof of the main result from Jiang et al. (2021), which is Theorem 4.1 of that work.

**Theorem 5** (Jiang et al. (2021))**.** *Let $\mathcal{A}$ be a stochastic training algorithm. If it is class-wise calibrated, then the error rate is equal to the expected disagreement rate between two trained networks:*

$$
\text{err}(\mathcal{A}) = \mathop{\mathbb{E}}_{h_1, h_2 \sim \mathcal{H}_{\mathcal{A}}^2, (x,y) \sim \mathcal{D}} [\mathbb{1}\{h_1(x) \neq h_2(x)\}]
$$

*Proof.* Let $q : \mathcal{X} \mapsto [0,1]^k$ be defined as in Section B.4. Then the laws of conditional expectation yield the following expression for the disagreement rate.

$$
\mathop{\mathbb{E}}_{h_1,h_2 \sim \mathcal{H}_{\mathcal{A}}^2,(x,y) \sim \mathcal{D}} [1\{h_1(x) \neq h_2(x)\}] = \mathop{\mathbb{E}}_{q} \left[ \mathop{\mathbb{E}}_{h_1,h_2,(x,y)} [1\{h_1(x) \neq h_2(x)\} \mid q(x) = q] \right]
$$

$$
= \mathop{\mathbb{E}}_{q} \left[ \mathop{\mathbb{E}}_{h_2,(x,y)} \left[ \mathop{\mathbb{E}}_{h_1} [1\{h_1(x) \neq h_2(x)\} \mid q] \mid q \right] \right]
$$

$$
= \mathop{\mathbb{E}}_{q} \left[ \mathop{\mathbb{E}}_{h_2,(x,y)} \left[ \sum_{c=1}^{k} q_c \cdot 1\{h_2(x) \neq c\} \mid q \right] \right]
$$

$$
= \mathop{\mathbb{E}}_{q} \left[ \mathop{\mathbb{E}}_{x,y} \left[ \mathop{\mathbb{E}}_{h_2} \left[ \sum_{c=1}^{k} q_c \cdot 1\{h_2(x) \neq c\} \mid q \right] \mid q \right] \right]
$$

$$
= \mathop{\mathbb{E}}_{q} \left[ \mathop{\mathbb{E}}_{x,y} \left[ \sum_{c=1}^{k} q_c(1 - q_c) \mid q \right] \right]
$$

$$
= \mathop{\mathbb{E}}_{q} \left[ \sum_{c=1}^{k} q_c(1 - q_c) \right].
$$

Each conversion of a conditional expectation over $\mathcal{H}_{\mathcal{A}}$ to a formula involving $q$ uses the assumption of class-wise calibration. This formula for the disagreement rate is the same one that we arrived at in Section B.4 for the error rate, so we have shown that the two are equal. □

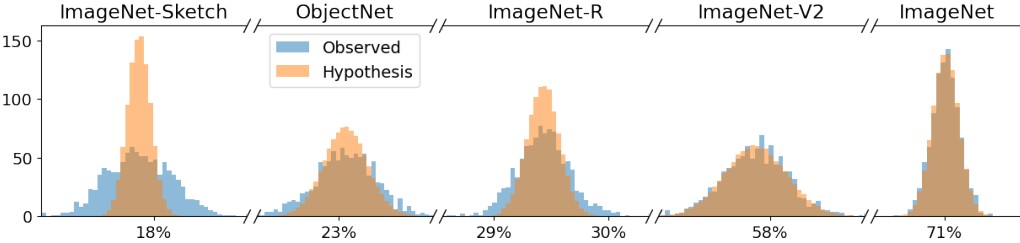

Figure 9: **Distribution shift produces excess distribution-wise variance between runs.** Across 1,000 runs of ImageNet training, both the ImageNet validation set and ImageNet-V2 have accuracy distributions close to that predicted by the independent errors assumption. On the other hand, distribution-shifted sets have significant excess variance, which indicates genuine differences between trained models, in light of Theorem 2.

## C  IMAGENET AND DISTRIBUTION SHIFTS

In this section we show that shifted distributions of test data have increased variance in their accuracy distributions between runs of training. We additionally confirm that the findings of Section 3 generalize to standard ImageNet training.

Our experiment is as follows. We independently train 1,000 ResNet-18s on ImageNet using a standard configuration (see Section A.2), attaining an average accuracy of 71.0%. We study the predictions of these networks on the ImageNet validation set, ImageNet-V2, and three distribution-shifted datasets.

We first look at the ImageNet validation set. In Figure 9 (rightmost) we observe that the true distribution closely matches the one predicted by independent errors. The observed standard deviation is 0.118%, but using Equation 3 we estimate distribution-wise variance to be $12\times$ smaller, at 0.034%. This value is close to what we found for CIFAR-10, confirming that both training scenarios adhere to our conclusions from Section 3.

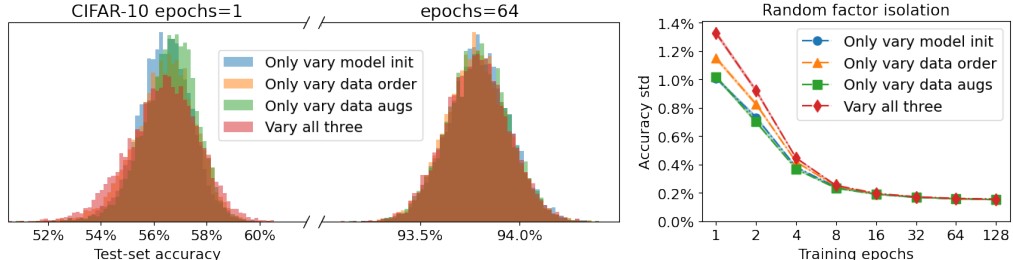

Figure 10: **One source of stochasticity suffices for long training.** When training for only 1 epoch, varying all three sources of randomness induces a standard deviation of 1.33% in test-set accuracy between runs, while each source alone induces 25-40% less variance. But when training for 64 epochs, varying any one source induces as much variance as all three together. Each distribution corresponds to 4,000 runs of training.

Next we consider ImageNet-V2 (Recht et al., 2019). This dataset is intended to have the same distribution of examples as ImageNet, and we demonstrate that its accuracy distribution has similar statistical properties as well. Specifically, we find that the distribution predicted by independent errors also closely matches the true distribution, and Equation 3 estimates a distribution-wise standard deviation of 0.071%, which is larger than what we found on the ImageNet validation set, but still relatively small. We note that the accuracy distribution for this dataset is wider, but assuming independent errors this can be explained simply by the fact that it has $5\times$ fewer examples than the ImageNet validation set.

By contrast, ImageNet-R (Hendrycks et al., 2021), ObjectNet (Barbu et al., 2019) and ImageNet-Sketch (Wang et al., 2019) have different statistical behavior compared to the first two datasets. These datasets are constructed to have shifted distributions relative to ImageNet. We find that their accuracy distributions have variance significantly in excess of that predicted by independent errors. We estimate using Equation 3 that these three test-sets have large distribution-wise standard deviations of 0.181%, 0.179%, and 0.257% respectively, indicating significant differences between runs of training.

We additionally investigate correlations between pairs of these five datasets (Figure 12). The strongest correlation is between ImageNet-R and ImageNet-Sketch, with $R^2 = 0.14$ ($p < 10^{-8}$). Manual inspection shows that both ImageNet-Sketch and ImageNet-R contain many sketch-like images, suggesting that similar features may induce correlation between distributions. All other pairs have $R^2 < 0.01$. For example, ImageNet-Sketch is decorrelated from ObjectNet, with $R^2 = 0.001$ ($p = 0.336$).

Overall, our findings suggest that training instability is in some sense a relative notion. ImageNet training is stable when evaluated on the main distribution, with a small standard deviation of 0.034% on the distribution of the ImageNet validation set. But it is unstable on shifted distributions, with ImageNet-Sketch having $58\times$ as much distribution-wise variance, at a standard deviation of 0.257%. This serves as a caveat to the title: from the perspective of the main training distribution, variance is harmless in that every trained network has almost the same performance, but from the perspective of shifted distributions, there are significant differences between runs.

# D    REPLICATION STUDY OF SUMMERS & DINNEEN (2021)

## D.1    THE THREE SOURCES OF RANDOMNESS

Training neural networks typically involves three sources of stochasticity, namely, model initialization, data ordering, and data augmentations. In this section we investigate how each of these sources contributes to the final variance between runs that we observe at the end of training.

We develop a CIFAR-10 training framework[3] that allows each source to be independently controlled by one of three different seeds. For example, when the data-augmentation seed is fixed and the data-order seed is varied, the set of augmented images seen by the network throughout training will

---

[3]https://github.com/KellerJordan/CIFAR10-isolated-rng

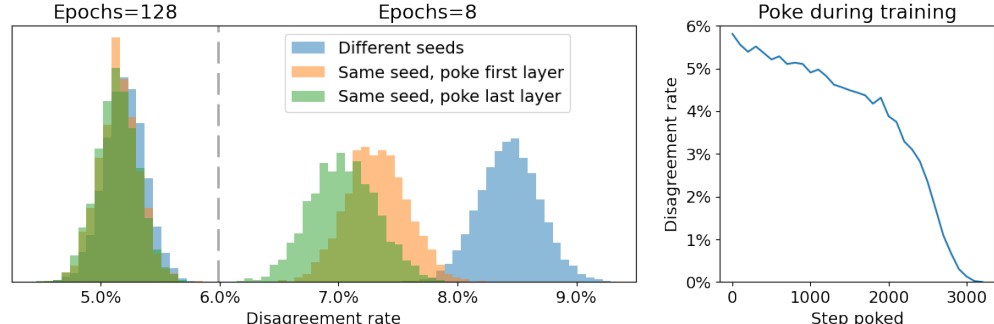

Figure 11: **Training has high sensitivity to initial conditions.** (Left:) For short trainings, pairs of runs which differ only by one network having been "poked" (*i.e.*, had a single weight changed slightly at initialization) disagree on 7.0-7.5% of predictions. Pairs of runs with fully different random seeds disagree more, on ∼8.5% of predictions. For long trainings, there is almost no difference. (Right:) The earlier a network is poked during the training process, the more its predictions will disagree with the network that trained unperturbed with the same random seed.

remain the same, but be presented in a different order. When all three seeds are fixed, training is deterministic, so that repeated runs produce the same network. Standard training is equivalent to allowing all three seeds to vary.

We fix two seeds, and vary just the third (*e.g.*, varying only the data order while keeping the model initialization and data augmentations fixed). Our naive intuition is that each factor contributes some part to the overall variance, so that this should decrease variance relative to the baseline of varying all three seeds.

Our results show that for short trainings of 1-16 epochs, this intuition is correct (Figure 10). For example, when training for 4 epochs, if we fix the data order and augmentations, while varying only the model initialization, then variance in test-set accuracy is reduced by 26%, with the standard deviation going from $0.45 \pm 0.01\%$ to $0.38 \pm 0.01\%$.

However, for longer trainings of 32 epochs or more, varying only one of the three random factors produces approximately the same variance as the baseline of varying all three. For example, across 8,000 runs of training for 64 epochs, varying just the model initialization (with data ordering and augmentation fixed) produces a standard deviation of 0.158%, almost the same as the baseline, which has 0.160%. At $n = 8,000$ this is not a statistically significant difference; it is possible that the true values are the same, or that they differ by a small amount. We conclude that for this training regime, any single random factor suffices to generate the full quantity of variance, rather than each factor contributing to overall variance.

### D.2 SENSITIVITY TO INITIAL CONDITIONS

In the previous section, we showed that when training to convergence, varying just the model initialization (or just the data ordering, or augmentations) produces approximately the same quantity of variance between runs as a baseline fully random setup. In this section we find that even varying a single weight at initialization suffices. Our findings replicate the work of Summers & Dinneen (2021), who reach similar conclusions.

Consider multiplying a single random weight in the network by 1.001. We call this "poking" the network. This is a tiny change; recent work in quantization (*e.g.*, Dettmers et al., 2022) implies that trained models can typically have *all* their weights modified more than this without losing accuracy.

Nevertheless, in Figure 11 we demonstrate that poking the network early in training produces a large difference in the final result. Our experiment is to run two trainings with the same random seed, but with one network being "poked" at some point during training. We measure the disagreement rate between the two networks, *i.e.*, the fraction of their test-set predictions that differ. For short trainings, poking induces much less disagreement than changing the random seed. But when training for 128 epochs, poking alone produces an average disagreement of 5.14%, barely less than the 5.19%

produced by using two entirely different random seeds. We have also observed that varying just the first batch of data, or the numerical precision of the first step (*e.g.*, fp16 vs. fp32) has a similar effect. We conclude that almost all variation between runs is not produced by specific sources of randomness like model initialization, data ordering, etc., but is instead intrinsic to the training process, which has extreme sensitivity to initial conditions.

# E ADDITIONAL FIGURES

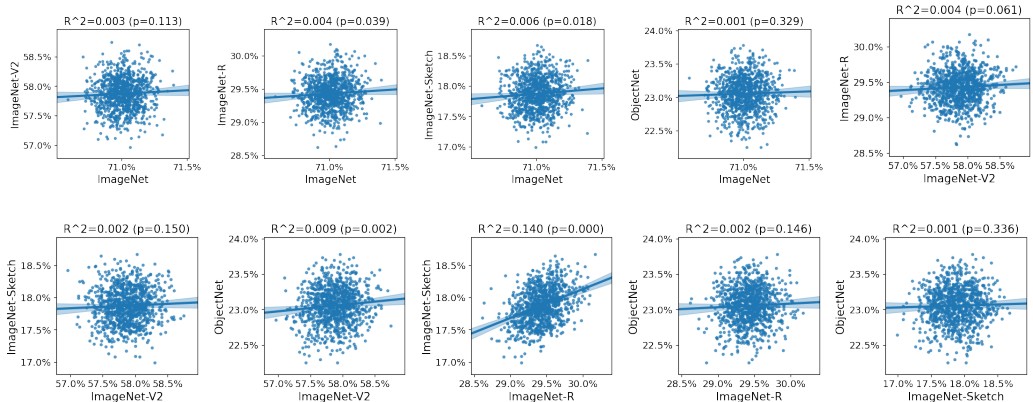

Figure 12: **Correlations between distribution shifts.** We visualize the accuracy values of 1,000 ResNets which were independently trained on ImageNet. Each network is evaluated on the ImageNet validation set, as well as four distribution-shift datasets (IN-V2, IN-R, IN-Sketch, and ObjectNet). We display the scatterplots of accuracy on each of the $\binom{5}{2}$ pairs. The following pairs had statistically significant correlations: (ImageNet-R, ImageNet), (ImageNet-Sketch, ImageNet), (ObjectNet, ImageNet-V2), and (ImageNet-Sketch, ImageNet-R). All but one pair have weak correlations with $R^2 < 0.01$. The strong correlation is between ImageNet-R and ImageNet-Sketch with $R^2 = 0.14$. We hypothesize that this is caused by the fact that both sets contain many sketch-like images, so that this pair has a) similar distributions and b) shifted distributions relative to the training set. We report two-sided p-values.

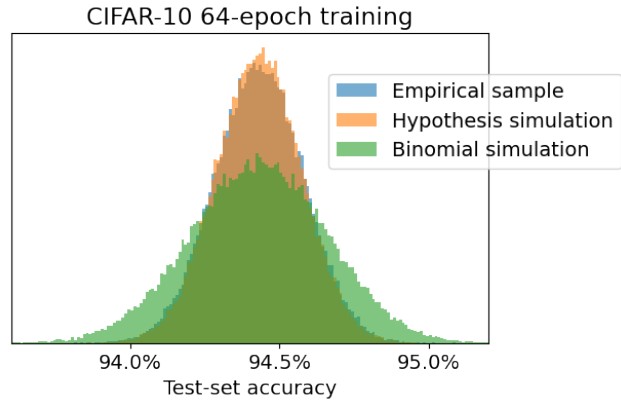

Figure 13: **The binomial approximation overestimates variance**. Compared to the empirical distribution of test-set accuracy, the binomial approximation predicts a distribution with too much variance. We use $p = 0.9441$ (the average accuracy) and $n = 10,000$ (the size of the test-set) to simulate $60,000$ samples from $\text{Binom}(n, p)$, which we find overestimates variance by a factor of $\approx 2.5\times$. In comparison, the independent errors assumption provides an accurate estimate.

| | | $P(C_1)P(C_2)$ | $P(C_1C_2)$ |
|---|---|---|---|
| | | 46.76% | 59.78% |
| | | 58.46% | 64.57% |
| | | 40.89% | 43.63% |
| | | 25.46% | 27.96% |
| | | 43.88% | 46.04% |

Figure 14: **Pairs with non-independent errors.** The first column is the product of the probability (over training stochasticity) that the trained network predicts the first example correctly, and the probability that it predicts the second example correctly. The second column is the probability that the trained network predicts both correctly at the same time. All probabilities are measured for our 64-epoch training configuration. The independent errors condition (Definition 1) would be if these two quantities were equal. Out of all $\binom{10,000}{2}$ pairs of examples in the CIFAR-10 test-set, only these five deviate by more than 2% from having independent errors.

