# OpenReview forum: "On the Variance of Neural Network Training with respect to Test Sets and Distributions"
_ICLR.cc/2024/Conference — ICLR 2024 poster_

### Official Review · Reviewer_Nr6n · 2023-10-24

**Soundness:** 3 good
**Presentation:** 3 good
**Contribution:** 3 good
**Rating:** 6
**Confidence:** 4

**Summary:**

In this work, the authors try to address the following important question: are marginal improvements in test accuracy (after a certain point) on benchmark datasets actually indicative of better models (with respect to the underlying data distribution)? Towards this end, the authors conduct a series of large-scale experiments on CIFAR-10 and ImageNet which show that trained model performance becomes uncorrelated on disjoint splits of test data. They also connect this variance in model performance with the notion of calibration of ensembles of models.

**Strengths:**

1. **Originality:** The experimental setups in this paper, as well as the connection introduced between model variance and calibration, are to the best of my knowledge new and contain several interesting ideas.
2. **Quality:** The authors have run extensive experiments to verify their hypotheses (at least within the context of the benchmarks they restrict themselves to), and their theoretical predictions track the experiments quite closely (for the appropriate regimes).
3. **Clarity:** Overall the paper is very easy to read, and justification/sufficient detail are provided for the experiments conducted in the paper.
4. **Significance:** This paper studies the important problem of how to assess a trained model's future test performance, and introduces useful formalisms for thinking about this problem. The main drawback here is that the paper's experiments and hypotheses are restricted to two image classification benchmarks (which are widespread - this is still a useful contribution), so it is tricky to assess how well the observations will generalize.

**Weaknesses:**

## Main Weaknesses
1. **Generalization of takeaways.** The practical takeaway provided in the paper in Section 3 is that marginal test performance improvements for well-trained models on CIFAR-10/ImageNet can be uncorrelated with performance on the underlying data distribution. I have some concerns with this claim (i.e. fixed test splits vs resampling), but even assuming this to be true - what can be said for more general machine learning (or even image classification) tasks? For example, in the experiments in Section 4, as the authors note there is a clear correlation between BERT-Large validation performance and test performance. I am not really sure how this should be interpreted in the context of the paper; while the analysis in the paper shows empirically that BERT-Large fine-tuning has more variance than BERT-Base, I don't see how the analysis in the paper would allow one to predict (without running many experiments) whether val performance will be correlated with test performance or not.
2. **Hyperparameter choices/sensitivity for results.** Of course it is not possible to be entirely comprehensive with respect to hyperparameters, but I have concerns with some choices made by the authors. Particularly, the training horizons considered are 0, 4, 16, and 64 epochs. We see in the experimental results a clear trend of decreasing variance as the training horizons are extended - do any of the results still hold for longer training horizons? I would anticipate that the independence of network predictions surely shrinks as training horizons are extended.

## Minor Comments
- In the proof of Lemma 1, the exponent should be inside the expectation. Additionally, it's probably helpful to justify the penultimate step (interchange of expectation) by saying that Fubini's Theorem applies.
- It would be better to include a definition of ECE in Section 3.4, or at least verbally describe it in the context of Hypothesis 2.

## Overall Recommendation
My concerns with the paper are mostly with the actual significance of the practical recommendations in the paper (as discussed above). That being said, I think the paper contains sufficiently many interesting ideas and experiments to possibly warrant acceptance (I am borderline because again, I'm not sure how well these ideas generalize). Thus, I recommend **weak accept**.

**Questions:**

- Is the test set split fixed? In other words, when the authors write "we split the test-set into two halves of 5000 examples each", does this mean there is a single fixed split of the test set? If so, this seems to not make sense in the context of the proposed formalism ($S \sim \mathcal{D}^n$), which considers randomly sampled test-sets.

---

> ### Author Response · Authors · 2023-11-21
> **Response to reviewer Nr6n**
>
> We thank the reviewer for their valuable comments, which give us an opportunity to clarify the generality of our results. We also thank the reviewer for noting the originality of our experiments and the significance of the subject being studied. We hope that these clarifications will satisfactorily address the reviewer’s concerns.
>
> Summary:
> * The result does generalize to longer training durations. In particular, Figure 5 shows that it generalizes to durations of 128 and 256 epochs, where the distribution-wise variance continues to be small.
> * The reviewer mentioned a concern that the takeaway did not generalize to finetuning BERT-Large, which we analyzed in Section 4.1. We comment that we included BERT-Large because it is a well-recognized case of pathological training instability [1, 2], so it is not surprising or problematic that the takeaway does not generalize to it.
> * Our results thus allow one to predict that for standard (not pathologically unstable) trainings which run to convergence, there will be little correlation between the performance on two splits of test data.
>
>
> **Clarification of the main experiment, regarding the fixed test-set split**
>
> The reviewer asked whether our test-set split is fixed. Yes, in Figure 2 we use a single fixed split of the test-set, into two halves.
>
> To say why this is the case, we’ll first give a summary of the main experiment, which proceeds as follows:
>
> We first generated a fixed split of the CIFAR-10 test-set into two halves of 5,000 examples each. Then we trained 60,000 models on the CIFAR-10 training set, and evaluated them all on both test splits.
>
> We found that the best-performing model on split 1 reached significantly higher performance than average - almost 95%, vs. an average of 94.2%. But this best model on split 1 didn’t do any better than average on split 2. In fact, we measured that the correlation between performance on the two splits is close to zero.
>
> Resampling the split does not change this result - the resulting correlation will always be near the same value.
>
> We also note that the mathematical analysis which proceeds in section 3.3 does not depend on using splits at all. Instead, the estimator given in Theorem 2 directly computes the average pairwise correlation.
>
> We hope that this clarifies how and why splits are used in our analysis, and shows that using a fixed split is not a problem.
>
>
> **Improvements to the presentation of the proof of Lemma 1**
>
> We have updated the draft to add extra parentheses inside of the expectation brackets, such that the square is moved inside. We also noted as suggested that Fubini’s theorem is what allows the interchange of expectations.
>
>
> **Generalization to longer training durations**
>
> The reviewer mentioned a concern that our results do not generalize to even longer durations, suggesting that perhaps when the training horizon is extended even further, the test-set variance may significantly continue to decrease, rather than plateauing as we have suggested.
>
> We would like to respond that both Figure 3 (right) and Figure 5 do present the results for training horizons all the way to 256 epochs, where it can be seen that the test-set accuracy does indeed plateau. From 64 to 256 epochs (which is roughly after the point of convergence, i.e. accuracy stops going up), we observe that test-set variance plateaus and examplewise predictions vary approximately independently between runs of training.
>
>
> **Generalization of takeaways**
>
> The reviewer mentioned that they have some concerns with our main claim, which is that for well-trained models on CIFAR-10/ImageNet, there is significant variance between runs in terms of performance on the test-set, but almost zero variance on the test-distribution.
>
> We hope that we have addressed the question about fixed test splits, and demonstrated that the claim does hold for standard CIFAR-10 and ImageNet trainings.
>
> The reviewer mentioned that their concern also stems from our study of BERT-Large, where indeed the claim does not hold, as there is a clear correlation between two splits of data.
>
> This is because BERT-Large is a case of pathological training instability, as has been studied in prior works [1, 2]. Any intervention which creates training instability will invalidate our claim by creating massive variance with respect to both the test-set and test-distribution. For example, our claim also would not generalize to trainings which use a huge learning rate, or a crazy network initialization which causes instability.
>
> Instead, our claim applies to standard trainings, which we directly confirmed for CIFAR-10 and ImageNet.
>
> We hope that this clarifies the generality of our takeaways.

---

> > ### Comment · Reviewer_Nr6n · 2023-11-21
> >
> > Thank you very much for the detailed response; my apologies for missing that Figures 3 and 5 covered longer training horizons. I have no further questions.

---

### Official Review · Reviewer_yu6g · 2023-10-28

**Soundness:** 3 good
**Presentation:** 2 fair
**Contribution:** 3 good
**Rating:** 5
**Confidence:** 4

**Summary:**

The authors analyze the variance between trained networks on the test set and find that minor variations are insignificant and explained by finite sample noise due to a finite test set. Further, they show that when considering the test distribution, the variations in validation set performance are not well correlated with the test distribution performance.

**Strengths:**

- The work formalizes what I think many people would ultimately suspect in that the minor variations between runs are insignificant when looked at through the larger perspective of the test distribution. Even though I think this would be expected, it is nice to see a work which empirically verifies it.

- The derived bounds are useful for estimating the expected variance between runs.

**Weaknesses:**

- The two main itemized contributions contain way too many points to be in a list. They are paragraph sized. I do not think something of that size should be presented in a list. I believe they either need to be broken up into smaller components, or discussed in plain text instead of a list.

- What is $n$ in theorem 3, 4, and 5? I suppose it should be the size of the dataset, but it would be nice to include this right by the theorems in order to avoid any possible confusion.

- I think it would be interesting to see the difference in Theorem 3 when varying the number of models in the ensemble. For instance, the bias in the estimate may be small, and a two model ensemble may give good results, or it may actually require quite a large ensemble in order to be close to correct. Can you add this experiment? I believe it can be done by only randomly choosing and expanding the ensemble of the models which are already trained.

- Many of the claims listed as main contributions are in the appendix. Even in the conclusion, the things in the appendix as main contributions which have been demonstrated even though there has been nothing said about them up until this point in the text. I do not think this is fair to the reader and it should be reorganized or rewritten such that this does not happen.

- In section 3.4 it says: “That is, if we let S ′ be the subset of test images which are classified by 30-40% of independently trained networks as “dog”, then approximately 30-40% of the images in S ′ really will be dogs.” I do not think that is how calibration is phrased in most calibration works. In fact, all networks may correctly classify the image as a dog (meaning that the highest predicted probability is the dog class), but the predicted probability (confidence) should equal the empirical distribution of the positive class in $S^\prime$, then the model is calibrated.

**Questions:**

- In figure 2, why is (odd examples) in parentheses? I cannot figure out what this means since there are only two splits and they seem to be uniformly random.

---

Ultimately, I like the findings of this work, but I find part of the presentation problematic as noted in the weaknesses section above. If these things can be fixed, I would probably raise my score.

---

> ### Author Response · Authors · 2023-11-21
> **Response to reviewer yu6g**
>
> We thank the reviewer for their valuable feedback, which has prompted us to improve the presentation of our results. We also thank the reviewer for noting that our derived bounds are useful for estimating the variance between runs, and prompting us (alongside reviewer Wzaj) to derive an additional new theorem with practical implications. We hope that these corrections and additions shall satisfactorily address the reviewer’s concerns, in which case we hope that the reviewer will raise their score.
>
> **Improvements to the presentation**
>
> We thank the reviewer for pointing out several problems with the presentation, which prompted us to make the following improvements to the draft:
>
> * We have converted the list of contributions in the introduction into two paragraphs.
> * We have clarified in the statements of Theorem 3-5 that $n$ refers to the size of the test-set.
> * Regarding the claims listed in the conclusion (regarding connections between variance and other phenomena), we have made it more explicit that two of them are presented only in the appendix. First, in the beginning of Section 4 we refer to the two additional experiments in the appendix. Second, in the conclusion we have added explicit pointers to the section that each claim is presented in. We regret that space issues precluded them being put directly in the main text.
>
> **Regarding the caption of odd vs even examples in Figure 2**
>
> We have added the following sentence to section 3.1, which describes the reason for this: “CIFAR-10 is already shuffled, so for convenience we simply use the odd and even-indexed examples as the two halves.” We have confirmed that if we use a newly selected random split instead, it makes no difference for the results.
>
> **Regarding the definition of calibration**
>
> We agree with the reviewer that our notion of calibration, called “ensemble-calibration” and referencing [1], is different from the standard version of calibration defined for single models.
>
> Indeed, the standard single-model calibration described typically *does not hold*, as neural networks tend to be overconfident [2]. On the other hand, the ensemble-calibration we describe *does* approximately hold, as observed in [1] and implicitly in [3].
>
> This is good because ensemble-calibration is mathematically what is needed to prove Theorem 3-5, whereas standard calibration is not useful.
>
> We hope this has clarified our usage of the notion of calibration.
>
>
> **Applying Theorem 4 to ensembles of two models**
>
> In the paper, we focused on the variance in performance between models that are members of large ensembles of >1,000 independently trained networks. The reviewer made the interesting suggestion that we should look at smaller ensembles of two networks.
>
> We found this suggestion interesting and have added a new theorem to the paper which derives the expected difference in accuracy between two models.
>
> This can be found as Theorem 6 in the appendix. We prove that for binary classification, the expected squared difference between the test-set accuracies of two independently trained models is equal to the average error divided by the size of the test set.
>
> We proved this under the assumptions of theorem 4, which are shown to approximately hold by the main experiments of the paper.
>
> We hope that this theorem will provide practitioners with a guideline for how much variance to expect to see between pairs of trained models. In particular, if two models are trained with different hyperparameters, and their accuracy differs by significantly more than the prediction of Theorem 6, then there may be a genuine difference between the quality of the two hyperparameters. Otherwise, there may be no statistically significant difference.
>
> [1] Lakshminarayanan, Balaji, Alexander Pritzel, and Charles Blundell. "Simple and scalable predictive uncertainty estimation using deep ensembles." Advances in neural information processing systems 30 (2017).
>
> [2] Guo, Chuan, et al. "On calibration of modern neural networks." International conference on machine learning. PMLR, 2017.
>
> [3] Nakkiran, Preetum, and Yamini Bansal. "Distributional generalization: A new kind of generalization." arXiv preprint arXiv:2009.08092 (2020).

---

> ### Comment · Reviewer_yu6g · 2023-11-23
>
> Thank you for the response.
>
> ---
>
> > The reviewer made the interesting suggestion that we should look at smaller ensembles of two networks.
>
> This is not what I suggested. As stated in the original review...
>
> >I think it would be interesting to see the difference in Theorem 3 when **varying the number of models in the ensemble**. For instance, the bias in the estimate may be small, and a two model ensemble may give good results, **or it may actually require quite a large ensemble** in order to be close to correct.
>
> I suggested to *vary* the number of models in the ensemble to analyze the bias/variance tradeoff of ensemble size.
>
> ---
>
> > We agree with the reviewer that our notion of calibration, called “ensemble-calibration” and referencing [1], is different from the standard version of calibration defined for single models.
>
> I do not think the paper referenced has a 'different from the standard' view of calibration. The paper referenced creates an equally weighted mixture of ensemble outputs
>
>  > **[from section 2.4 of [1]]**: For classification, this corresponds to averaging the predicted probabilities.
>
> Therefore calibration is measured by the average of the predicted probabilities and not as you described in the paper.
>
> ---

---

> > ### Author Response · Authors · 2023-11-23
> > **Response to reviewer yu6g**
> >
> > Thank you for your reply.
> >
> > If the new theorem does not match what you had in mind - for the case that the number of models in the ensemble equals two - then we would request a clarification.
> >
> > What do you mean by varying the number of models in the ensemble? Do you mean in order to compute the ECE, as we defined it? Or something else?
> >
> > Empirically, the ECE (as we have defined it, which is the way it must be defined for the math to work out) is small. We therefore intend theorem 4 to be the main practical tool, which we empirically confirmed is approximately accurate in Figure 6.
> >
> > Regarding the difference between our definition of calibration and that of Lakshminarayanan et al. 2017:
> > You are right that there is a difference. We did not realize this until now. Our version would correspond to averaging the one-hot predictions from all models in the ensemble, rather than the predicted probabilities. Or equivalently, using temperature zero in the softmax.
> >
> > It empirically turns out that even with this modification, which is necessary for the math to work out, approximate calibration observed still holds. Perhaps this is a new result, albeit a minor one.
> >
> > Would you agree that our notion of calibration is equivalent to Lakshminarayanan et al. 2017, modified by taking one-hot or temperature=0? If we can make this explicit in the text, would you view it as fixing the problem?

---

> ### Author Response · Authors · 2023-11-23
> **Response to reviewer yu6g**
>
> We have updated the draft (in section 3.4) in order to correctly reference the notion of calibration that was previously explored in Lakshminarayanan et al. 2017.
>
> We thank the reviewer for pointing out this issue, and regret not investigating it more closely before our first reply.
>
> We believe that all of the presentational issues brought up by the reviewer have now been addressed. We thank the reviewer for helping us improve the presentation of the paper.

---

### Official Review · Reviewer_Wzaj · 2023-10-29

**Soundness:** 3 good
**Presentation:** 3 good
**Contribution:** 3 good
**Rating:** 8
**Confidence:** 4

**Summary:**

This paper delves into the variance observed in test-set performance across repeated runs of neural network training, a phenomenon that has raised concerns about hyperparameter comparison and training reproducibility. Key findings of the study are:

- Although there's significant variance in test-set performance, the variation on the underlying test-distributions (like CIFAR-10 and ImageNet) is minimal. This suggests that in practical applications, the variance might not be as concerning as assumed.

- The study puts forth an "examplewise independence" hypothesis, suggesting that a network's correct prediction for a given test example is akin to a biased coin flip and is independent of its predictions on other examples.

- The paper states that prior works have noted that predictions from ensembles of networks, trained repeatedly, are typically calibrated. They argue that this calibration inevitably leads to some variance in test-set accuracy. For binary classifications, they provide a formula to determine this variance.

- Other observations include the reduction in distribution-wise variance with longer training durations, the correlation between optimal learning rate and absence of excess distribution-wise variance, the effect of data augmentation in reducing variance, and the increase in variance when test-sets differ from the training set.

**Strengths:**

This paper covers an interesting topic: the variation of test accuracy over random seeds (stochasticity of deep learning experiments). It tells the test accuracy on the whole test set from the test accuracy on the test distribution (random subsets of test sets). An important conclusion is that accuracy gain in the whole test set from random seeds does not generalize to the whole test distribution. Another interesting conclusion is that the result of trained model on each example can be approximated by binomial distribution with biased flip probability.

Overall, I find this paper interesting and novel. The paper is easy to understand.

**Weaknesses:**

Althouth I enjoy reading this paper, the analysis is mainly posterior: we can only get these insights after running the same algorithm for hundreds of runs. As far as I can see, the insights from this paper can tell us if a training algorithm is good despite the impact of random seed, but it cannot tell if a trained model is better than another despite the impact of random seed. If this paper can further achieve the latter application, I would be more than happy to raise my score.

Another concern is how well does the conclusion itself generalize to other settings. In Figure 1, it seems the variance becomes smaller as training goes on. But I think it is caused by the learning rate schedule: the authors say that they "always linearly ramp the learning rate down to zero by the end of training", so it is expected that the accuracy has less variation in the end.

**Questions:**

- How does the conclusion generalize to other training setting or other network architecture (like Transformers)?

- How can the insights of random seeds help practical development? E.g. if they can be used to tell the quality of individual model, to separate the accuracy into random part and true accuracy.

---

> ### Author Response · Authors · 2023-11-21
> **Respond to reviewer Wzaj**
>
> We thank the reviewer for their valuable feedback, and for noting that our paper is interesting and novel along with being easy to understand. The reviewer's feedback that the paper was lacking in practically useful results motivated us to make the addition of a new theorem which can be used as an applied tool. We hope that our updated draft and response will satisfactorily address the reviewer’s concerns, in which case we hope that the reviewer will raise their score.
>
> Response summary:
> * To improve the practical applicability of our results, we have added a new theorem (Theorem 6 in Section C of the updated draft) which predicts the expected squared difference in accuracy between two runs of training. For example, this can be used as a tool to check whether a pair of hyperparameters yields significantly differing performance.
> * Regarding generalization to other settings, we clarify that even for our short trainings we anneal the learning rate to zero. That is, for our four-epoch trainings, we ramp the learning rate to zero by the end of the fourth epoch. We also have preliminary evidence that our results extend to an NLP/transformer setting.
>
> **Practical application of our analysis**
>
> The reviewer expressed a concern that although our main analysis yields important conclusions, it is mainly posterior, the only implication for practice being that the practitioner should beware that variation on the test-set likely will not translate to real performance gains on the test-distribution.
>
> However we submit that Theorem 4, and the newly added variant which is Theorem 6, provide a significant practical tool. In Theorem 4, we prove that under certain assumptions which we find to hold empirically, it is the case that for binary classification we can expect the variance between runs to be precisely the average error divided by twice the number of test examples. This prediction was found to be empirically quite accurate in Figure 6.
>
> As a corollary, the expected squared difference between the performance of two trained models is twice this, i.e., the average error divided by the number of test examples (Theorem 6). We suggest that this can be used as a tool to test the significance of hyperparameter ablations: Suppose we train two models with hyperparameters A and B and measure their squared difference in performance. Then if this value is less than the expectation given by Theorem 6, then there may be no statistically significant difference between A and B. On the other hand, if it is much more, then this may suggest a genuine difference in hyperparameter quality.
>
> We acknowledge that it remains an open question how to extend this result to k-way classification beyond k=2, and how to derive precise concentration inequalities giving statistical significance values, rather than the simple predicted expectation given by Theorem 6. But we hope that the theorem resolves the reviewer’s main practical concern, given that it already constitutes a significant practical application of our analysis.
>
>
> **Clarification of the learning rate schedule**
>
> > In Figure 1, it seems the variance becomes smaller as training goes on. But I think it is caused by the learning rate schedule: the authors say that they "always linearly ramp the learning rate down to zero by the end of training", so it is expected that the accuracy has less variation in the end.
>
> We have updated the draft (in section 2) to clarify that this means that, even for the four-epoch duration, we ramp the learning rate down to zero by the end of the four epochs. Thus, none of our trainings, no matter how short, end with a nonzero learning rate. We hope that we have correctly understood the reviewer’s concern.
>
> We note that if the learning rate were not ramped all the way down, then indeed the variation would be greatly increased for the shorter trainings. But this isn’t what we did.
>
>
> **Generalization to transformers**
>
> In Section 4.1, we confirm that for a canonical transformer-encoder finetuning scenario (BERT-base finetuned on MRPC), our results do empirically generalize, with test-distribution variance being small relative to test-set variance. For BERT-Large finetuning our results do not generalize, but this seems to be explained by this being a widely recognized case of pathological training instability [1, 2].
>
> Thus, we have at least preliminary evidence that our results generalize to this quite different other setting. We agree with the reviewer that, for example, it is an interesting direction for future work to extend our study to transformers trained from scratch. (e.g. LLM pretrainings)
>
>
> [1] Mosbach, Marius, Maksym Andriushchenko, and Dietrich Klakow. "On the stability of fine-tuning bert: Misconceptions, explanations, and strong baselines." arXiv preprint arXiv:2006.04884 (2020).
>
> [2] Dodge, Jesse, et al. "Fine-tuning pretrained language models: Weight initializations, data orders, and early stopping." arXiv preprint arXiv:2002.06305 (2020).

---

> > ### Comment · Reviewer_Wzaj · 2023-11-22
> >
> > Thanks for the reply. The extension of Theorem 6 is too restricted, and the generalization results are not very good. I will keep my initial score.
> >
> > I don't understand the clarification of the learning rate schedule. My thoughts are that, the observed behavior might be correlated with learning rate schedule.

---

### Official Review · Reviewer_G5ah · 2023-11-01

**Soundness:** 3 good
**Presentation:** 3 good
**Contribution:** 2 fair
**Rating:** 5
**Confidence:** 3

**Summary:**

This paper shows that the variance for true test distribution among multiple training runs is actually smaller than those observed from the test dataset. They also proposed a statistical assumption and derived an estimator to estimate the stand deviation of the test distribution.

**Strengths:**

The paper is easy to follow. The paper provides many empirical analysis to support their points and also include some theoretical analysis.

**Weaknesses:**

Since most parts of the paper are based on the empirical analysis, it might be better the have the experimental setup in the main paper including all the hyperparameters and all the models in those 60000. However, it might be important to include results for different architectures since some of the models may have better reproductions. It might be interesting to have more use cases for getting the variances of the true test distribution.

**Questions:**

Have we tried different architectures? Do they have similar results? I am curious do we really need to get the true variance? For example, if for all the models, we have a similar trend between variance from test-set and test-distribution, then why do we need to get that?

Do we have some cases that the variance among the test distribution gives us new insight? For example, does flatness of the minimizer related with this?

---

> ### Author Response · Authors · 2023-11-21
> **Response to reviewer G5ah**
>
> We thank the reviewer for their valuable feedback, and for noting that our paper is easy to follow. We provide the following response, which we hope will satisfactorily address the reviewer’s concerns, in which case we hope that the reviewer will raise their score.
>
> Regarding the concerns raised, we have the following summary:
>
> First we confirm that there is **no simple trend relating test-set and test-distribution variance**, making our in-depth analysis necessary.
> We are also happy to confirm that the other datasets and architectures that we did study in the paper do have similar results.
> Finally, we are happy to report that our analysis does yield new insights relating variance to other phenomena in deep learning: for example, section 4.2 studies the relationship to data augmentation and section D.1 studies the relationship to the learning rate. We agree that the flatness of minimizer is an interesting future direction.
>
> ### Hyperparameter details
>
> We thank the reviewer for pointing out that it would be better to have full hyperparameters in the main text, and have therefore added full detail for our 240,000 total runs of training on CIFAR-10 into Section 2 (Setup).
>
> ### On whether there exists simple trend connecting test-set and test-distribution variance
>
> The reviewer expressed a concern regarding the possibility that the relationship between variance on the test-distribution and the test-set may follow a simple trend, making it less interesting to study.
>
> We can confirm that no such trend exists, using the following evidence:
>
> As our first example, we compare the 16-epoch and 64-epoch training durations on CIFAR-10. These have similar test-set variance, but very different test-distribution variance.
>
> In particular, as shown in Figure 5, the 16-epoch training duration has an observed test-set std of 0.187%, which is 1.28x the observed test-set std of the 64-epoch trainings, which is 0.146%. So in terms of *test-set* variance, these two training configurations appear similar. But their estimated variance on the *test-distribution* is significantly different - the 16 epoch duration has 0.09%, while the 64-epoch duration has almost 3x less true variance, having 0.034%.
>
> So the trend is certainly not a linear one. It is not even monotonic either. For another example, training on 80% less training data for 64 epochs (figure 7) causes more test-set variance than training on full data for 16 epochs (figure 3) – but the latter causes more distribution-wise variance.
>
> ### New insights into connection between variance and other phenomena
>
> The reviewer asked whether studying the test-distribution variance can give any new insights, for example possibly connecting with flatness of minimizer. We agree that flatness of minimizer is an interesting future direction of study.
>
> We point out that our paper contains the following **connections between variance and other phenomena in deep learning**, which may be of interest:
> * The inclusion of data-augmentation is crucial for minimizing distribution-wise variance, moreso than one would expect given only its impact on error rate. (Section 3.2)
> * The optimal learning rate is the largest one that does not induce excess distribution-wise variance. (Appendix D.1 - unfortunately we couldn’t include this in the main text due to space constraints)
>
> ### Other architectures
>
> Regarding studying variance between runs of training using other neural network architectures, we point out that we studied finetuning the BERT (transformer-encoder) architecture in Section 4.1. We additionally replicated our results for ImageNet using a larger ResNet in Appendix D.2.

---

### Meta-Review · Area_Chair_wuef · 2023-12-05

**Metareview:**

This paper analyzes the variance between runs of neural network training, finding that it is basically harmless and in any case inevitable. The reviewers found the experiments thorough and revealing, and there was general agreement that the results were interesting and I think they will be appreciated by the ICLR community. Some of the weaknesses highlighted in the discussion phase include aspects of unclear presentation and there were some questions about the practical implications. The author response helped mitigate these concerns, and as such I recommend acceptance.

**Justification For Why Not Higher Score:**

While the authors have highlighted some of the practical implications of the work, the main takeaways are more along the lines of "when to quit tuning," which, while important, are perhaps not game-changing.

**Justification For Why Not Lower Score:**

The insights are novel, the experiments are thorough, and the analysis will be interesting for the ICLR community.

---

### Decision · Program_Chairs · 2024-01-16

Accept (poster)